# Decidual natural killer cells dysfunction is caused by IDO downregulation in dMDSCs with *Toxoplasma gondii* infection
Yu Wang[1,3], Xiaoyue Zhao[2,3], Zhidan Li[1,3], Wenxiao Wang[1], Yuzhu Jiang[1], Haixia Zhang[1], Xianbing Liu[1], Yushan Ren[1], Xiaoyan Xu[1] & Xuemei Hu [1]✉

Myeloid-derived suppressor cells (MDSCs) play a crucial role in maintaining maternal-fetal tolerance by expressing some immune-suppressive molecules, such as indoleamine 2,3-dioxygenase (IDO). *Toxoplasma gondii* (*T. gondii*) infection can break the immune microenvironment of maternal-fetal interface, resulting in adverse pregnancy outcomes. However, whether *T. gondii* affects IDO expression in dMDSCs and the molecular mechanism of its effect are still unclear. Here we show, the mRNA level of IDO is increased but the protein level decreased in infected dMDSCs. Mechanistically, the upregulation of transcriptional levels of IDO in dMDSCs is regulated through STAT3/p52-RelB pathway and the decrease of IDO expression is due to its degradation caused by increased SOCS3 after *T. gondii* infection. In vivo, the adverse pregnancy outcomes of IDO$^{-/-}$ infected mice are more severe than those of wide-type infected mice and obviously improved after exogenous kynurenine treatment. Also, the reduction of IDO in dMDSCs induced by *T. gondii* infection results in the downregulation of TGF-β and IL-10 expression in dNK cells regulated through Kyn/AhR/SP1 signal pathway, eventually leading to the dysfunction of dNK cells and contributing the occurrence of adverse pregnancy outcomes. This study reveals a novel molecular mechanism in adverse pregnancy outcome induced by *T. gondii* infection.

*T. gondii* is an obligate intracellular protozoan parasite that can infect any nucleated cells. *T. gondii* infection is almost asymptomatic in healthy individuals. However, in special populations, such as pregnant women, it can lead to abortion, stillbirth, or congenital malformations (microcephaly and hydrocephalus) and other severe complications[1,2]. Our previous studies demonstrated that the disruption of the immune microenvironment at the maternal-fetal interface contributed to adverse pregnancy outcomes induced by *T. gondii* infection[3–6]. The maternal-fetal immune system comprises diverse interactive decidual immune cells and their related molecules. Several studies demonstrated that maternal-fetal tolerance was sustained by various decidual cells, including decidual natural killer (dNK) cells, decidual macrophages (dMφ), decidual regulatory T cells (dTregs), and decidual dendritic cells (dDCs)[7]. More recently, the novel myeloid-derived suppressor cells (MDSCs) have drawn increasing attention due to their remarkable immune suppressive properties[8]. Some researchers have demonstrated that dMDSCs play an essential role in sustaining human pregnancy by expressing immune-suppressive molecules, such as indoleamine 2,3-dioxygenase (IDO), arginase 1 (Arg-1), inducible nitric oxide

synthase (iNOS) and interleukin-10 (IL-10)[9,10]. IDO is an intracellular heme-containing enzyme that exhibits biological effects by catalyzing L-tryptophan into kynurenine (Kyn), which mainly expressed in professional antigen-presenting cells (APCs)[11]. A study showed that the expression and activity of IDO in decidual tissues of women with unexplained recurrent spontaneous abortion were significantly reduced[12]. Our group found that IDO was downregulated in dDCs after *T. gondii* infection during early pregnancy[13,14]. However, it was unclear whether *T. gondii* affected IDO expression in dMDSCs and what its mechanism was.

It was reported that the signal transducer and activator of transcription 3 (STAT3)/non-classical nuclear factor kappa-B (NF-κB) pathway could directly induce IDO expression in DC of peripheral blood and MDSCs of infiltrated breast cancer[15,16]. The NF-κB transcription factor family comprises five members: p50, p52, p65, c-Rel and RelB[17,18]. After various extracellular and intracellular stimuli activate the non-classical NF-κB pathway, the phosphorylation of IκB kinase (IkappaB kinase α, IKKα) directly phosphorylates p100. The presence of the p100-RelB dimer in the cytoplasm does not exhibit transcriptional activity. However, upon

[1]Department of Immunology, Binzhou Medical University, Yantai 264003 Shandong, PR China. [2]Department of Clinical Psychology, Yantai Affiliated hospital of Binzhou Medial University, Yantai 264100 Shandong, PR China. [3]These authors contributed equally: Yu Wang, Xiaoyue Zhao, Zhidan Li. ✉e-mail: xue-mei-hu@163.com

phosphorylation of p100, it undergoes ubiquitination and is subsequently processed into p52 by the proteasome. The p52-RelB dimer with transcriptional capability is formed and transported into the nucleus to initiate the transcription of target genes[19,20]. In MDSCs of breast cancer infiltration, phosphorylated STAT3 can activate the non-canonical NF-κB signaling pathway, leading to the phosphorylation of p100 and the nuclear translocation of p52-RelB. P52-RelB dimers can directly bind to the IDO promoter and modulate IDO expression[16]. One study reported that a *T. gondii* kinase protein could directly bind with STAT3 and induce its phosphorylation[21]. However, it needs to be explored whether the change of IDO expression in dMDSCs after *T. gondii* infection is regulated by the STAT3/p52-RelB pathway. In early pregnancy, the expression of IDO in chorionic villus and decidual tissues were negatively correlated with suppressors of cytokine signaling 3 (SOCS3)[22]. SOCS3 is a protein factor that plays a negative regulatory role in the cytokine signal transduction pathway, and the expression levels of SOCS3 in cells are low or almost negligible under normal circumstances[23]. Certain factors can stimulate the production of SOCS3 in cells, leading to the degradation of the SOCS3-bound target protein by proteasomes[23,24]. At the post-translation level, SOCS3 could bind to the phosphorylated immunoreceptor tyrosine-based inhibitory motifs (ITIMs) of IDO, resulting in ubiquitinylation and degradation[25,26]. However, it is unclear whether IDO expression in dMDSCs during *T. gondii* infection is regulated via the STAT3/p52-RelB signaling pathway and SOCS3 protein.

IDO is a crucial enzyme in tryptophan metabolism that catalyzes tryptophan (Trp) to produce Kyn. The Kyn content can be used as an indicator to assess IDO activity[11]. In the maternal-fetal interface, IDO can create a tryptophan-deficient microenvironment, inhibiting T cell proliferation, inducing immune tolerance of Treg cells and affecting the proliferation and function of dNK cells[27]. All decidual immune cells work together to maintain maternal-fetal tolerance[8]. Depleting dMDSCs at the maternal-fetal interface can upregulate the cytotoxicity of dNK cells, potentially leading to pregnancy failures[28]. However, the impact of the interaction between dMDSCs and dNK cells on the maternal-fetal immune imbalance caused by *T. gondii* infection has not been documented. The production of Kyn by IDO in dMDSCs can play an immunomodulatory role by combining with its natural ligand, the aromatic hydrocarbon receptor (AhR), which belongs to a class of ligand-activated transcription factors primarily found in the cytoplasm[11]. After activation, AhR binds to an aryl hydrocarbon receptor nuclear translocator (ARNT) to form a dimer. This complex is then translocated into the nucleus, where it triggers the expression of a cascade of downstream target genes[29]. AhR is highly expressed in dNK cells and is involved in the normal development and growth of placental blood vessels[30]. A study reported that recurrent abortion was related to the abnormal expression of AhR[29]. The reports showed that the upregulation of IDO in trophoblast cells enhanced the inhibition of dNK cytotoxicity through the IDO/Kyn/AhR pathway[31,32]. However, it is necessary to further investigate whether the changes in IDO in dMDSCs after *T. gondii* infection could result in the dysfunction of dNK cells through the IDO/Kyn/AhR pathway.

Human dNK cells are dominated by a unique CD56[bright]CD16[−] phenotype and express interleukin-10 (IL-10) and transforming growth factor-β (TGF-β) to maintain maternal-fetal immune tolerance and promote fetal development[33]. Our previous studies demonstrated that decreased levels of IL-10 and TGF-β contributed to adverse pregnancy outcomes in case of *T. gondii* infection[34–36]. However, the molecular mechanisms involving the reduction of IL-10 and TGF-β in dNK cells remain unclear. Researches have shown that decreased AhR levels can downregulate the expression of IL-10 and TGF-β[37,38]. The study also found that the AhR/ARNT dimer recruited the nuclear transcription factor specificity protein 1 (SP1) by binding to the zinc finger structural domain of SP1, forming the AhR/ARNT-SP1 complex[39]. Next, the nuclear transcription factor SP1 binds to the promoters, enhancing the transcription and expression of IL-10 and TGF-β[40–42]. However, whether the alteration of IDO in dMDSCs after *T. gondii* infection regulates the production of IL-10 and TGF-β in dNK cells through the Kyn/AhR/SP1 pathway, ultimately leading to the dysfunction of dNK cells needs to be explored.

This study investigated the molecular mechanisms involving IDO changes in dMDSCs after *T. gondii* infection and how they further regulate the functions of dNK cells. Our results will help elucidate the molecular immune mechanisms of adverse pregnancy outcomes caused by *T. gondii* infection.

## Results

### IDO⁻ᐟ⁻ pregnant mice have more severe abnormal pregnancy outcomes after *T. gondii* infection than those of infected WT pregnant mice

Decidual MDSCs are essential for sustaining maternal-fetal immune tolerance and ensuring pregnancy success[9]. As a negative immunomodulator at the maternal-fetal interface, IDO in dMDSCs plays a crucial immunoregulatory function during pregnancy[10]. Therefore, this study investigated the role of IDO of dMDSCs on adverse pregnancy outcomes resulting from *T. gondii* infection. *T. gondii*-infected IDO⁻ᐟ⁻ pregnant mouse model was successfully established to compare adverse pregnancy outcomes between IDO⁻ᐟ⁻ pregnant mice and infected WT pregnant mice. Interestingly, the adverse pregnancy outcomes of infected IDO⁻ᐟ⁻ pregnant mice were more severe than those of infected WT mice, including more pronounced arching of the back, severe placental hemorrhage, and fetal mice were almost unformed (Fig. 1a–c). Moreover, in infected IDO⁻ᐟ⁻ mice, placental and fetal weights were all significantly decreased, and the percentage of stillbirths was noticeably increased (Fig. 1d). These results indicated that deficiency in IDO might be essential in the adverse pregnancy outcomes caused by *T. gondii* infection.

### IDO expression levels in dMDSCs decreased after *T. gondii* infection

To investigate the change of IDO expression in dMDSCs after *T. gondii* infection, human dMDSCs were purified from decidual tissues and then infected with *T. gondii*. The qPCR results showed that the mRNA levels of IDO were significantly increased in the infected group compared to those in the uninfected group (Fig. 2a). However, the expression levels of IDO in the infected group were significantly decreased by western blot and flow cytometry analysis (Fig. 2b, c). In addition, similar results were obtained in vivo, the IDO expression levels in dMDSCs of infected mice were significantly lower than those of uninfected mice (Fig. 2d).

### *T. gondii* infection promoted STAT3 phosphorylation and activated the STAT3/p52-RelB pathway

In breast cancer-derived MDSCs, STAT3-induced activation of the non-canonical NF-κB signaling pathway promoted an immunosuppressive microenvironment by regulating the expression of IDO[15]. To investigate whether *T. gondii* infection could promote STAT3 phosphorylation and activate the STAT3/p52-RelB pathway, we examined the levels of p-STAT3 and downstream members of the STAT3/NF-κB pathway (p-IKKα, p-p100, p52, RelB) in dMDSCs after *T. gondii* infection. In vitro, the mRNA levels of p52, RelB and IDO in human dMDSCs were increased after *T. gondii* infection by qPCR (Fig. 3a). The levels of p-STAT3, p-IKKα and p-p100 in human dMDSCs increased after *T. gondii* infection by western blot. Also, the results indicated that *T. gondii* infection could upregulate the protein expression of p52 and RelB in the nucleus of human dMDSCs (Fig. 3b, c). However, the protein level of IDO in human dMDSCs after *T. gondii* infection was significantly decreased (Fig. 3b, c). Consistent with the results of in vitro, we found that the expression of p-STAT3 in dMDSCs of infected mice was increased compared to that in uninfected mice by flow cytometry (Fig. 3d). These results suggested that *T. gondii* infection could promote STAT3 phosphorylation and activate the STAT3/p52-RelB signaling pathway.

### *T. gondii* infection regulated the expression of IDO in dMDSCs through the STAT3/p52-RelB pathway

A detailed mechanism was explored to understand whether the changes of IDO in dMDSCs by *T. gondii* infection are regulated by the STAT3/p52-

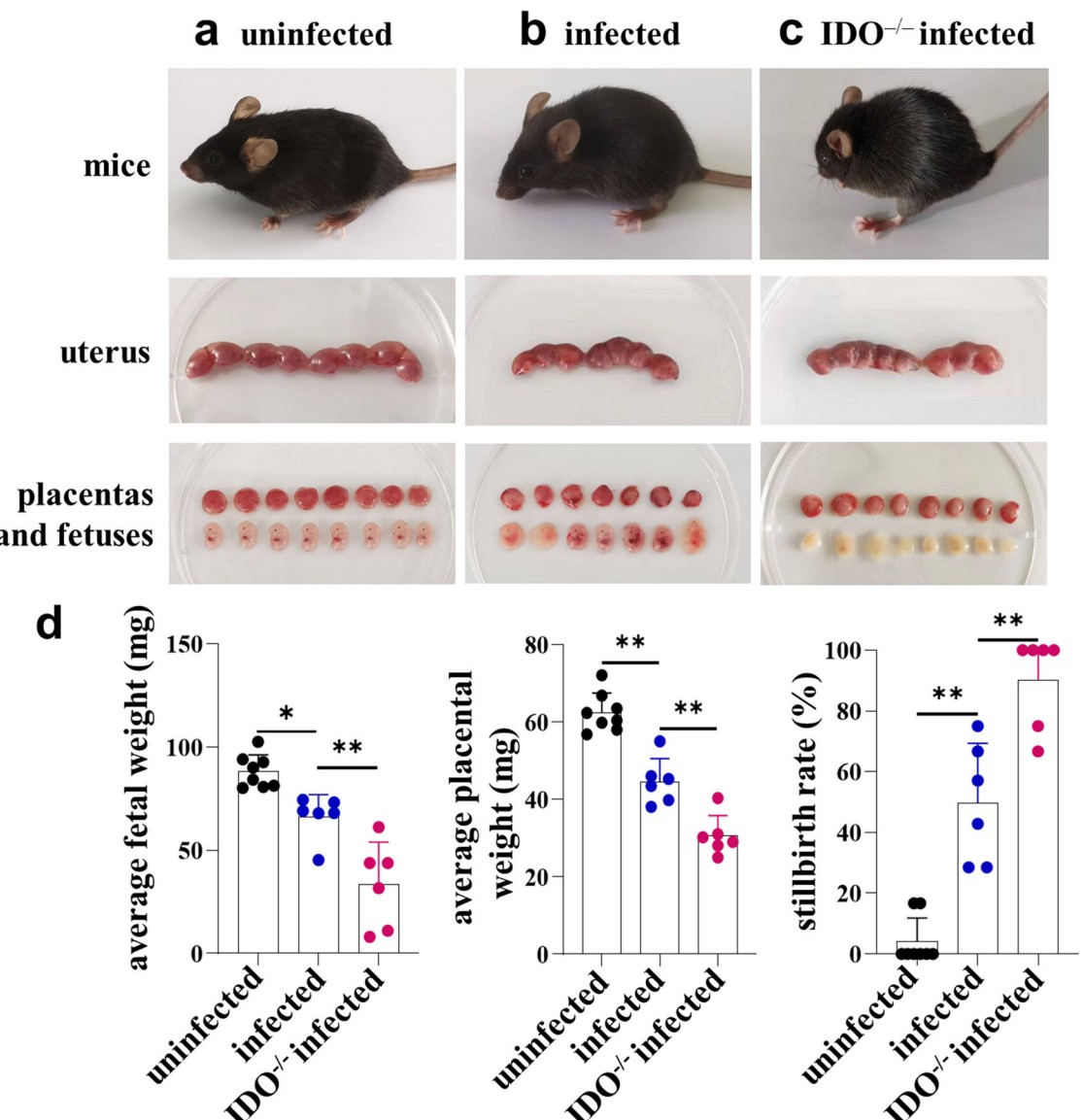

**Fig. 1 | Effects of IDO on abnormal pregnancy outcomes caused by *T. gondii* infection in pregnant mice.** The uterus, developmental status of placentas and fetuses in the WT uninfected mice (**a**), WT infected mice (**b**) and IDO$^{-/-}$ infected mice (**c**). **d** The weight of the placentas and fetuses and the stillbirth rate in WT uninfected (*n* = 8), WT infected (*n* = 6), and IDO$^{-/-}$ infected mice (*n* = 6). Data are presented as the mean ± SD, \**P* < 0.05, \*\**P* < 0.01, mice samples in each group were assayed individually by one-way ANOVA.

RelB pathway. The human purified dMDSCs were infected and treated with a STAT3 phosphorylation inhibitor (stattic) or a p52-RelB nuclear translocation inhibitor (SN52), respectively. Interestingly, after inhibiting STAT3 phosphorylation during *T. gondii* infection, the mRNA levels of IDO decreased by qPCR (Fig. 4a). Subsequently, western blot analysis revealed that the protein expressions of p-STAT3, p-IKKα, p-p100 and IDO all decreased. Additionally, the nuclear translocations of p52 and RelB were also reduced (Fig. 4b). Furthermore, after inhibiting the nuclear translocations of p52 and RelB with SN52 during *T. gondii* infection, the mRNA levels of IDO were downregulated (Fig. 4a). This result was consistent with the trend of changes in IDO protein levels detected by western blot (Fig. 4c). In vivo, infected mice were intraperitoneally injected with a STAT3 inhibitor (JSI-124). Consistent with the results of in vitro, the expression of IDO in dMDSCs of mice was significantly decreased after treatment with JSI-124 (Fig. 4d). Taken together, these results suggested that *T. gondii* infection regulated the expression of IDO in dMDSCs via the STAT3/p52-RelB pathway.

## SOCS3 expression in human dMDSCs was significantly increased during *T. gondii* infection

It was found that the expression of IDO at the maternal-fetal interface was negatively correlated with SOCS3[22]. Some studies reported that the combination of SOCS3 with IDO can enhance the ubiquitination and degradation of IDO at the post-translational level[23,24]. Both mRNA and protein levels of SOCS3 in infected human dMDSCs were significantly increased compared to those in uninfected human dMDSCs, as determined by qPCR and western blot analysis, respectively (Fig. 5a, b). These results indicated that the mRNA levels of IDO increased, but the protein levels decreased after *T. gondii* infection. This might be related to the increased expression of SOCS3, resulting in IDO degradation at the post-translational level. Further, CO-IP experiments were conducted to determine if IDO could interact with SOCS3 during *T. gondii* infection. The results showed that the interaction between IDO and SOCS3 was enhanced after *T. gondii* infection (Fig. 5c). Our results suggested that the increased expression of SOCS3 during infection might promote the ubiquitination degradation of IDO and ultimately lead to the decrease of IDO protein levels.

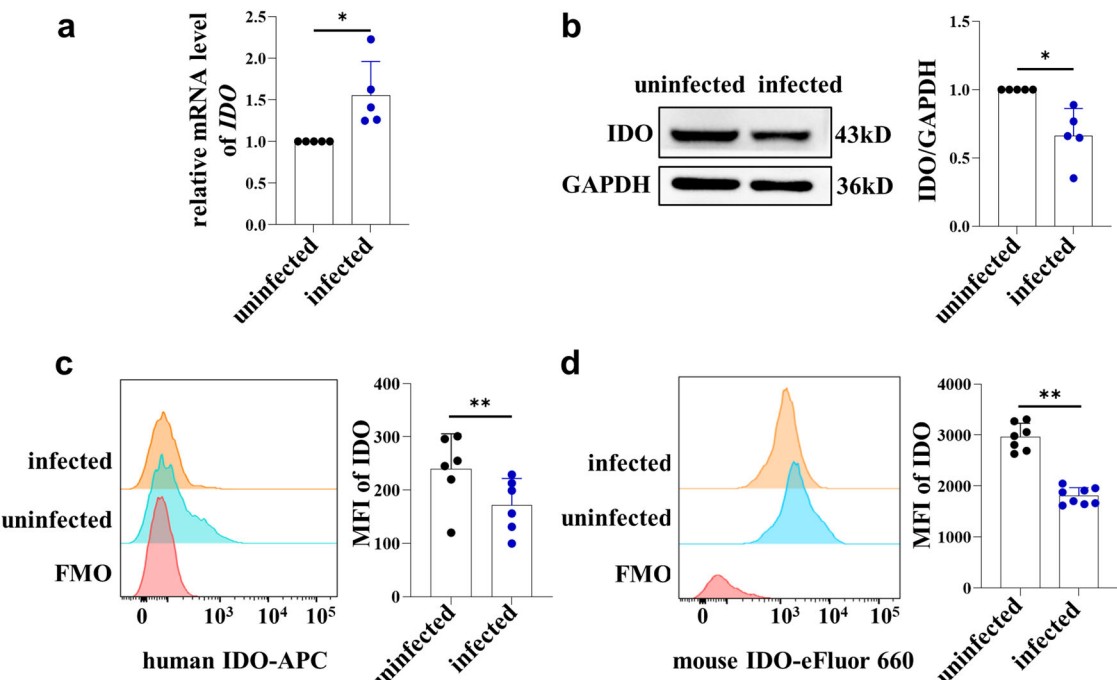

**Fig. 2 | Expression of IDO in human and mouse dMDSCs between uninfected group and infected group. a** The IDO mRNA levels in human dMDSCs in uninfected and infected groups were analyzed by qPCR (data represent the mean ± SD from five independent experiments). The IDO protein expression in human dMDSCs was examined in uninfected and infected groups by western blot (**b**) (data represent the mean ± SD from five independent experiments) and flow cytometry (**c**)

(data represent the mean ± SD from 6 human samples). **d** The IDO expression in mouse dMDSCs in uninfected ($n = 7$) and infected groups ($n = 8$) was analyzed by flow cytometry. Data are presented as the mean ± SD, *$P < 0.05$, **$P < 0.01$, human samples in each group assayed individually by the paired $t$-test and mice samples in each group assayed individually by the unpaired $t$-test. MFI mean fluorescence intensity.

## Reduction of IDO in dMDSCs after *T. gondii* infection might affect the expression of TGF-β and IL-10 in dNK cells

IDO can degrade tryptophan to create kynurenine, and the enzyme activity of IDO can be measured by the content of Kyn[11]. In order to investigate whether the downregulation of IDO induced by infection could affect the production of Kyn, the levels of Kyn in the supernatant of human dMDSCs were detected by HPLC. The results showed that the Kyn levels in the supernatant of the infected group were significantly lower than those of the uninfected group, and the Kyn levels further decreased after treatment with the IDO inhibitor (1-MT) (Fig. 6a). In vivo, the levels of Kyn in the placental supernatant of infected mice were reduced compared to those of uninfected mice, and the Kyn levels were lower in infected IDO$^{-/-}$ mice than in infected mice (Fig. 6b). To investigate whether the downregulation of IDO in dMDSCs induced by *T. gondii* infection affected the function of maternal-fetal tolerance of dNK cells, purified human dMDSCs were divided into uninfected and infected groups, and were co-cultured with dNK cells using a transwell system in vitro. The results showed that after co-culturing with infected dMDSCs, the expression levels of TGF-β and IL-10 in dNK cells were significantly decreased compared to those with the uninfected dMDSCs (Fig. 6c). In vivo, the expression levels of TGF-β in infected mouse dNK cells were significantly decreased compared to those of uninfected mice. Furthermore, the expression levels were further decreased in infected IDO$^{-/-}$ mice as analyzed by flow cytometry (Fig. 6d). In addition, to investigate the effect of Kyn on the interaction between dMDSCs and dNK cells, dMDSCs were treated with an IDO inhibitor 1-MT and exogenous Kyn, respectively, in the co-culture system. Interestingly, the results showed that the expression levels of TGF-β and IL-10 in dNK cells were decreased with 1-MT treatment, while they were increased when treated with exogenous Kyn (Fig. 6e). These results suggested that the decrease of IDO in dMDSCs after *T. gondii* infection could affect the expression levels of TGF-β and IL-10 in dNK cells. To further verify that the adverse pregnancy outcomes of IDO$^{-/-}$ mice induced by *T. gondii* infection are related to Kyn, exogenous Kyn was used to treat infected IDO$^{-/-}$ pregnant mice. To our

surprise, the adverse pregnancy outcomes of IDO$^{-/-}$ infected mice after treatment with exogenous Kyn were significantly improved, showing less wrinkled fur, decreased placental hemorrhage and necrosis, and reduced proportion of abnormal fetuses (Fig. 6f). Additionally, the average weight of the placenta and fetal mice were increased, and the proportion of stillbirths were decreased (Fig. 6g). Meanwhile, the expression levels of TGF-β in dNK cells of infected IDO$^{-/-}$ mice treated with Kyn were significantly increased (Fig. 6h). These results suggested that the decrease of IDO in dMDSCs with *T. gondii* infection could affect the function of maternal-fetal tolerance of dNK cells mediated by kyn.

## Reduction of IDO in dMDSCs after *T. gondii* infection regulated the expression of TGF-β and IL-10 in dNK cells through Kyn/AhR/SP1 pathway

Kyn is an endogenous ligand of AhR and can promote the expression of AhR[11]. A study found that the AhR/ARNT dimer recruited the nuclear transcription factor SP1 by binding to the zinc finger structural domain of SP1, forming the AhR/ARNT-SP1 complex[39]. Next, SP1 binds to the promoter, enhancing the transcription and expression of IL-10 and TGF-β[40–42]. To explore the molecular mechanism by which the reduction of IDO in dMDSCs after *T. gondii* infection further regulate the function of maternal-fetal tolerance of dNK cells, the expression levels of AhR and SP1 in dNK cells of the co-culture system were detected by western blot. The results showed that the expression levels of AhR and SP1 were significantly decreased in human dNK cells after co-culturing with infected dMDSCs compared to those with the uninfected dMDSCs (Fig. 7a). Furthermore, the expression levels of AhR and SP1 were decreased after treatment with the IDO inhibitor 1-MT, but they were increased in human dNK cells treated with exogenous Kyn in the co-culture system (Fig. 7b). Consistent with the results of in vitro, the expression levels of AhR in mouse dNK cells from infected mice were significantly decreased compared to those of uninfected mice and further decreased in infected IDO$^{-/-}$ mice analyzed by flow cytometry (Fig. 7c). Meanwhile, after treatment with Kyn, the AhR

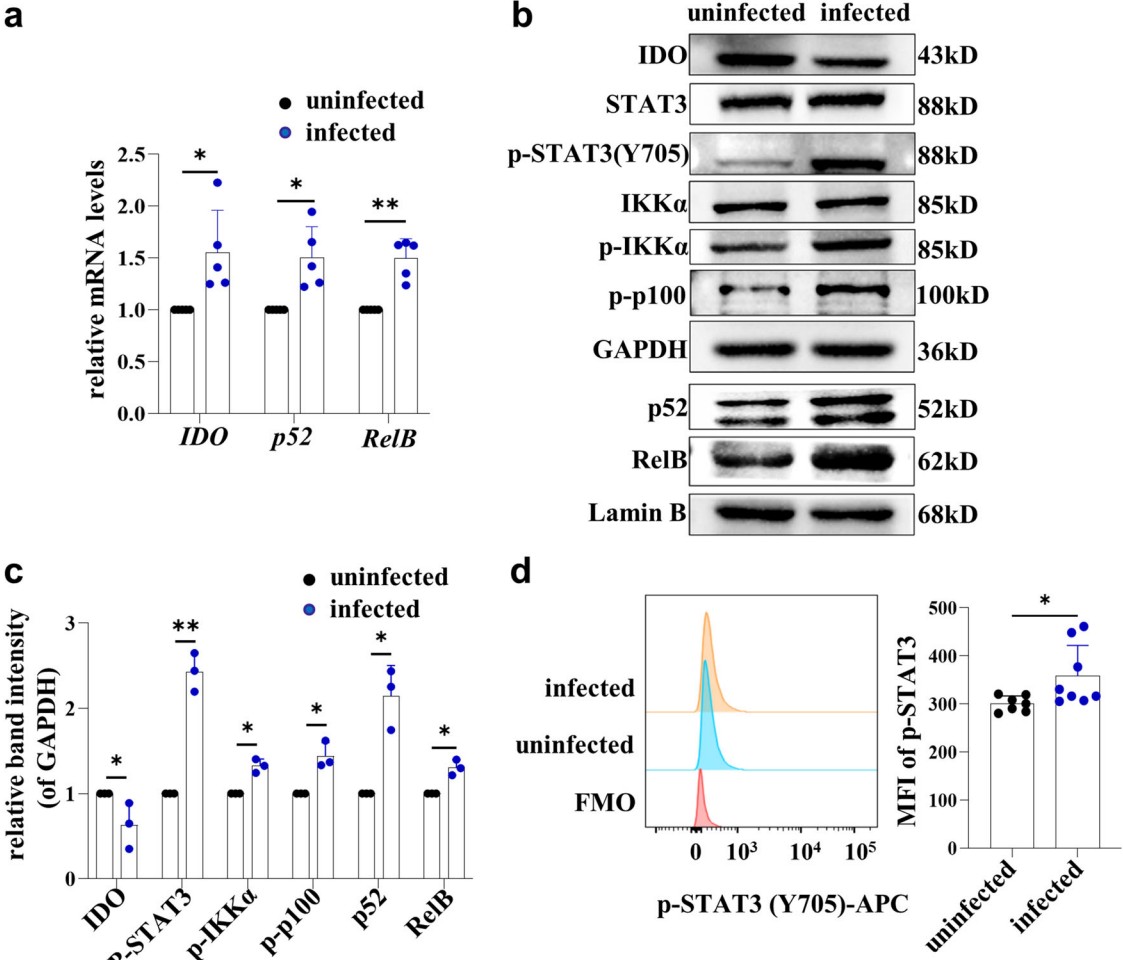

**Fig. 3 | *T. gondii* infection promoted STAT3 phosphorylation and activated the STAT3/p52-RelB pathway. a** The expression of p52, RelB and IDO in purified human dMDSCs of uninfected and infected groups were examined by qPCR (data represent the mean ± SD from five independent experiments). **b** The protein levels of IDO, STAT3, p-STAT3 (Y705), IKKα, p-IKKα, p-p100 in purified human dMDSCs, p52 and RelB in nucleus of dMDSCs of uninfected and infected groups were determined by western blot. **c** The statistical analysis of IDO, p-STAT3 (Y705), p-IKKα, p-p100, p52 and RelB in protein levels from infected groups and uninfected human dMDSCs by western blot. (data represent the mean ± SD from three independent experiments). **d** The expression levels of p-STAT3 in mouse dMDSCs were examined by flow cytometry in uninfected ($n = 7$) and infected groups ($n = 8$) and the results were analyzed by statistical analysis. Data are presented as the mean ± SD, *$P < 0.05$, **$P < 0.01$, human samples in each group assayed individually by the paired *t*-test and mice samples in each group assayed individually by the unpaired *t*-test. MFI: mean fluorescence intensity.

expression in dNK cells of infected IDO$^{-/-}$ pregnant mice were significantly increased. (Fig. 7d). These results suggested that IDO in dMDSCs may regulate the expression of TGF-β and IL-10 in dNK cells through the Kyn/AhR/SP1 pathway after *T. gondii* infection. To further explore whether the expression of TGF-β and IL-10 in dNK cells were regulated through the Kyn/AhR/SP1 pathway, AhR inhibitor (CH-223191) or SP1 inhibitor (Plicamycin) was used to treat human dNK cells in a co-culture system with dMDSCs, respectively. The results showed that the expression of AhR, SP1, TGF-β and IL-10 were decreased after treatment with CH-223191 (Fig. 7e). Meantime, there was no obvious change in the expression of AhR after treatment with Plicamycin, but the expression of SP1, TGF-β and IL-10 decreased substantially (Fig. 7f). Therefore, these results demonstrated that the reduction of IDO in dMDSCs after *T. gondii* infection could regulate the expression of TGF-β and IL-10 in dNK cells through the Kyn/AhR/SP1 signaling pathway.

### Proposed mechanistic model
*T. gondii* infection promoted STAT3 phosphorylation and upregulated the transcriptional levels of IDO in dMDSCs through the STAT3/p52-RelB pathway. However, at the post-translation level, an increased expression of SOCS3 during infection could bind to the phosphorylated ITIMs of IDO,

leading to its ubiquitination and degradation, ultimately resulting in decreased IDO expression. Then the production of Kyn was decreased as a result of the downregulation of IDO in dMDSCs after *T. gondii* infection and its ligand AhR was also decreased. The decreased AhR in dNK cells could reduce the recruitment of the nuclear transcription factor SP1 and regulate the expression levels of TGF-β and IL-10 in dNK cells via the Kyn/AhR/SP1 pathway (Fig. 8).

### Discussion
At the maternal-fetal interface, various decidual immune cells, immune molecules, and secreted cytokines create a tolerant immune microenvironment that is crucial for sustaining a healthy pregnancy[7]. Several studies found that the occurrence of unexplained recurrent abortion is closely related to the disruption of dynamic immune balance at the maternal-fetal interface[43,44]. *T. gondii* is an obligate intracellular protozoan parasite that triggers a robust immune response at the maternal-fetal interface when it infects during early pregnancy[45]. It could cause severe pregnancy complications, such as abortion, stillbirth, anencephaly, and congenital toxoplasmosis[1,2,46]. Our previous studies reported that abnormal pregnancy outcomes associated with *T. gondii* infection were attributed to the dysfunction of decidual immune cells resulting from the abnormal

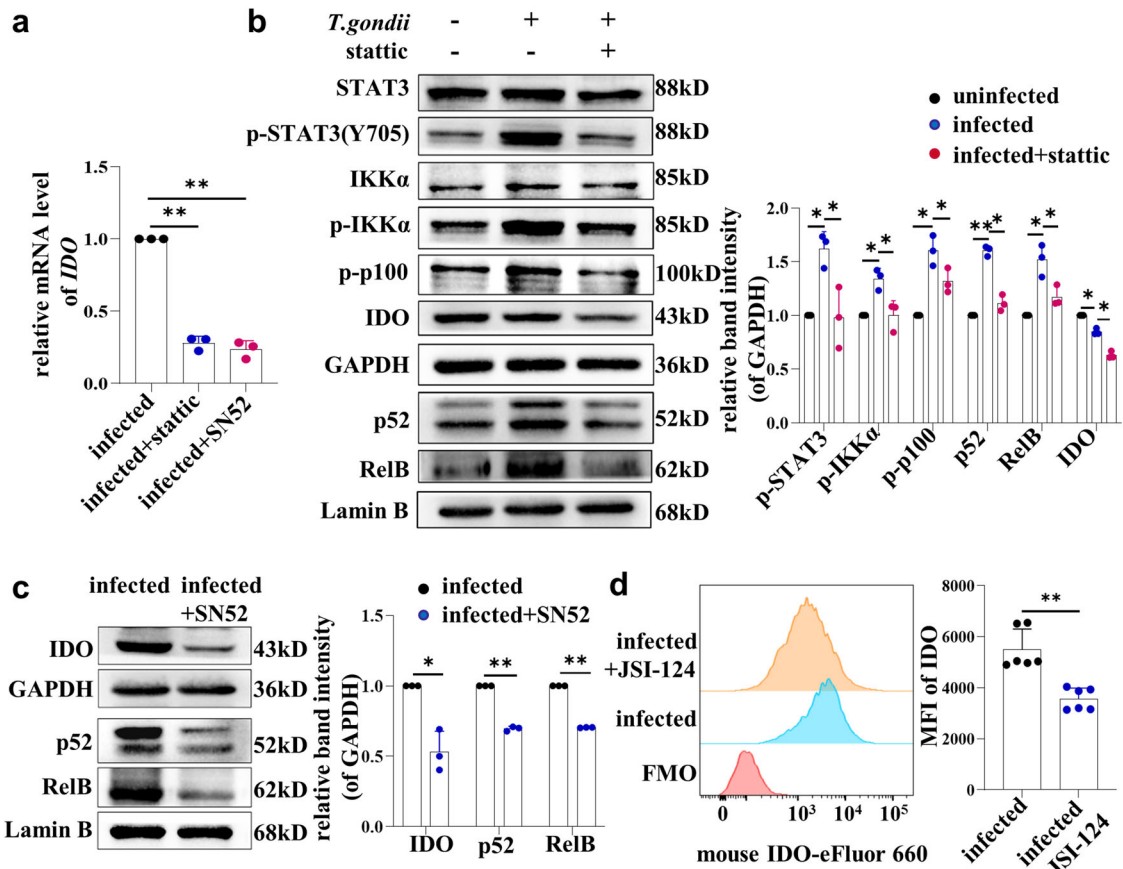

**Fig. 4 | *T. gondii* infection regulated IDO expression through STAT3/p52-RelB pathway. a** The mRNA levels of IDO in human dMDSCs of infected group, infected with static treated group and infected with SN52 treated group were detected by qPCR (data represent the mean ± SD from three independent experiments by one-way ANOVA). **b** The expression changes of STAT3, p-STAT3 (Y705), IKKα, p-IKKα, p-p100, IDO and the expression of p52 and RelB of nucleus in purifed human dMDSCs of the uninfected, infected and infected with static treated groups were determined by western blot (data represent the mean ± SD from three independent

experiments by one-way ANOVA). **c** The expression changes of IDO in human dMDSCs and the expression of p52 and RelB in nucleus of dMDSCs from the infected and infected with SN52 treated groups were determined by western blot (data represent the mean ± SD from three independent experiments by the paired *t*-test). **d** The IDO expression in mouse dMDSCs of infected and infected with JSI-124 groups were examined by flow cytometry (*n* = 6). Data are presented as the mean ± SD, *P < 0.05, **P < 0.01, unpaired *t*-test. MFI mean fluorescence intensity.

expression of functional membrane molecules and abnormal production of cytokines in dNK cells, dMDSCs, dMφ, dDCs and dTregs[3–6,47,48]. Decidual MDSCs have drawn increased attention due to their outstanding immune suppressive properties, which play an important role in sustaining human pregnancy[8,9]. Decidual MDSCs express some immuno-suppressive molecules, such as IDO, Arg-1 and IL-10[10]. Some studies have found that the number of dMDSCs in patients with unexplained recurrent abortion was decreased, and cell apoptosis occurred[49,50]. IDO, a negative immunomodulatory molecule, is highly expressed in dMDSCs and is initially believed to be involved in the negative regulation of T-cell-mediated immune responses by inhibiting proliferation. This plays a crucial immunosuppressive role in maintaining maternal-fetal tolerance[51]. A study showed that the expression and activity of IDO in decidual tissues of women with unexplained recurrent spontaneous abortion were significantly reduced[12]. These results indicated that IDO expression may be conducive to maintaining maternal-fetal tolerance. However, the alterations in IDO expression in dMDSCs following *T. gondii* infection and their potential association with adverse pregnancy outcomes require further investigation.

In the present study, a *T. gondii*-infected IDO[−/−] pregnant mouse model was successfully established. Interestingly, the adverse pregnancy outcomes of infected IDO[−/−] mice were more severe than those of infected WT mice, with more stillbirths and resorbed fetuses, as well as smaller placental and fetal sizes. These results indicated that IDO deficiency is essential in adverse pregnancy outcomes caused by *T. gondii* infection. To investigate the

potential impact of *T. gondii* infection on IDO expression in dMDSCs, human purified dMDSCs were infected with *T. gondii* in vitro. Results showed that both mRNA and protein levels of IDO in dMDSCs were all significantly changed after infection. Furthermore, similar results were obtained in vivo, the expression of IDO in dMDSCs was downregulated in *T. gondii*-infected mice compared to the uninfected mice. However, the mechanism underlying the changes in IDO in dMDSCs induced by *T. gondii* infection remains unclear.

It was reported that the STAT3/p52-RelB pathway could directly induce IDO expression in DCs of peripheral blood and MDSCs of infiltrated in breast cancer[15,16]. The phosphorylation of STAT3 in MDSCs infiltrated by breast cancer can induce the activation of the non-classical NF-kB signaling pathway and promote the formation of the p52-RelB dimer. This dimer can directly bind to the IDO promoter and regulate IDO expression[16]. However, it remains unclear whether the altered IDO expression levels in dMDSCs after *T. gondii* infection are related to the STAT3/p52-RelB pathway. In this study, we aimed to investigate whether *T. gondii* infection could enhance STAT3 phosphorylation and activate the STAT3/p52-RelB pathway. To achieve this, purified human dMDSCs were infected with *T. gondii* in vitro, and the model of pregnant mice infected with *T. gondii* were established in vivo. We found that the expression levels of p-STAT3 and the downstream members of the STAT3/p52-RelB pathway (p-IKKα, p-p100, p52 and RelB) in human dMDSCs were all increased. Meanwhile, the expression of p-STAT3 in infected mouse dMDSCs was increased. This indicated that

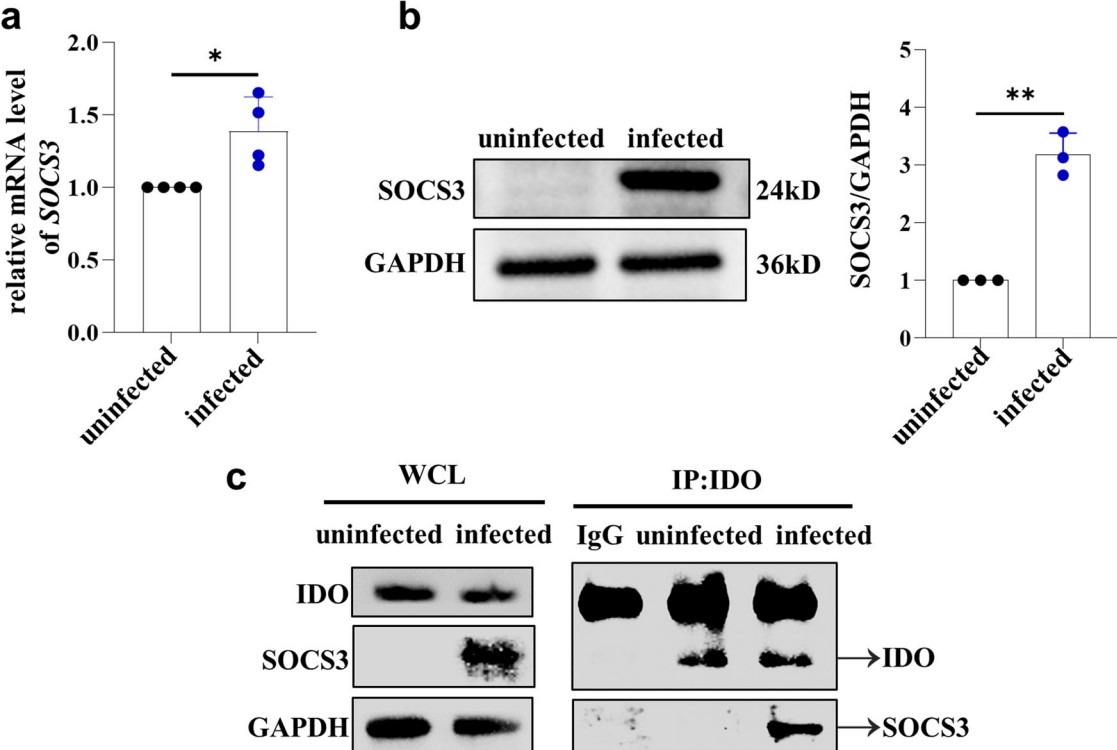

**Fig. 5 | Expression of SOCS3 in human dMDSCs after *T. gondii* infection and the interaction between SOSC3 and IDO. a** SOCS3 mRNA levels in human dMDSCs between uninfected and infected groups were examined by qPCR (data represent the mean ± SD from four independent experiments). **b** SOCS3 expression in human dMDSCs of uninfected and infected groups were examined by western blot (data represent the mean ± SD from three independent experiments). **c** The expressions of IDO and SOCS3 in IP group and WCL group were detected by western blot when we used IDO antibody to immunoprecipitate SOCS3. Data are presented as the mean ± SD, *$P < 0.05$, **$P < 0.01$, human samples in each group assayed individually by the paired *t*-test.

IDO changes in dMDSCs induced by *T. gondii* infection may be attributed to the activation of the STAT3/p52-RelB pathway. To further explore the molecular mechanism, human dMDSCs and pregnant mice were infected with *T. gondii*. They were then treated with a STAT3 inhibitor (stattic) or a p52-RelB nuclear translocation inhibitor (SN52) in vitro. Additionally, they were injected with a STAT3 inhibitor (JSI-124) *i.p.* in vivo. As anticipated, the levels of p-STAT3 in human dMDSCs were significantly downregulated in the infected group treated with a STAT3 inhibitor compared to the infected group. The nuclear translocation of p52 and RelB, the mRNA and protein levels of IDO were significantly reduced. These results demonstrated that STAT3 phosphorylation induced by *T. gondii* infection could regulate the expression of p52-RelB and IDO. After inhibiting the nuclear translocation of p52 and RelB with SN52, the mRNA and protein levels of IDO were downregulated. In vivo, infected mice were *i.p.* injected with JSI-124, the expression of IDO in mouse dMDSCs was significantly decreased, consistent with in vitro results. Therefore, our data explicitly showed that the expression of IDO in dMDSCs was regulated through the STAT3/p52-RelB pathway during *T. gondii* infection.

Unexpectedly, we found that the IDO expression in infected human dMDSCs was significantly increased at the mRNA level, but significantly decreased at protein level. A study reported that the expression of IDO in chorionic villus and decidual tissues were negatively correlated with SOCS3 during early pregnancy[22]. Other studies found that SOCS3 combined with the phosphorylated ITIMs of IDO could enhance the ubiquitination and degradation of IDO at the post-translation level[25,26]. So, whether the increased in IDO of infected human dMDSCs at the mRNA level but decreased at the protein level was due to degradation at the post-translation level resulting from changes in SOCS3 needs to be explored. Interestingly, we found that both mRNA and protein levels of SOCS3 in infected human dMDSCs were significantly increased. Next, the interaction between IDO and SOCS3 was explored by using the IDO IP antibody to capture the target proteins. The CO-IP results revealed that SOCS3 was present in the target proteins analyzed by western blotting following *T. gondii* infection. Therefore, our results indicated that SOCS3 could interact with IDO during infection. These results indicated that the decreased of IDO in protein level was due to the degradation at the post-translation level, resulting from an increase in SOCS3 expression induced by *T. gondii* infection. However, how the parasite interacts with dMDSCs or its effector molecules directly to induce the increase of IDO in translation and the reduction of IDO in protein level, the detailed mechanism still needs further exploration.

Studies have proved that WT mice infected with *T. gondii* in early pregnancy can cause serious adverse pregnancy outcomes. These outcomes are attributed to the dysfunction of immune cells, such as dMDSCs[47,48], dNK cells[3,52], dMφ[4,53] and dDCs[5,14] at the maternal-fetal interface. The present study showed that IDO expression was significantly decreased in infected dMDSCs, and the adverse pregnancy outcomes of IDO[-/-] pregnant mice were notably severe. It is indicated that the changes of IDO induced by *T. gondii* infection might play an important role in the adverse pregnancy outcomes. However, the immune molecular mechanism by which the downregulation of IDO expression caused by *T. gondii* infection contributed to the adverse pregnancy outcomes still needs to be clarified.

A study found that IDO can create a tryptophan-deficient micro-environment in the maternal-fetal interface, inhibiting T cell proliferation, inducing immune tolerance of Treg cells, and affecting the proliferation and function of dNK cells[27]. Another study discovered that depletion of dMDSCs at the maternal-fetal interface could enhance the cytotoxicity of dNK cells, resulting in pregnancy failures[28]. Human dNK cells account for about 70% of all decidual immune cells and highly express IL-10 and TGF-β to maintain maternal-fetal immune tolerance and promote fetal development[33]. In order to investigate whether IDO downregulation in dMDSCs induced by *T. gondii* infection could lead to dysfunction of dNK cells, the purified infected human dMDSCs were co-cultured with dNK cells.

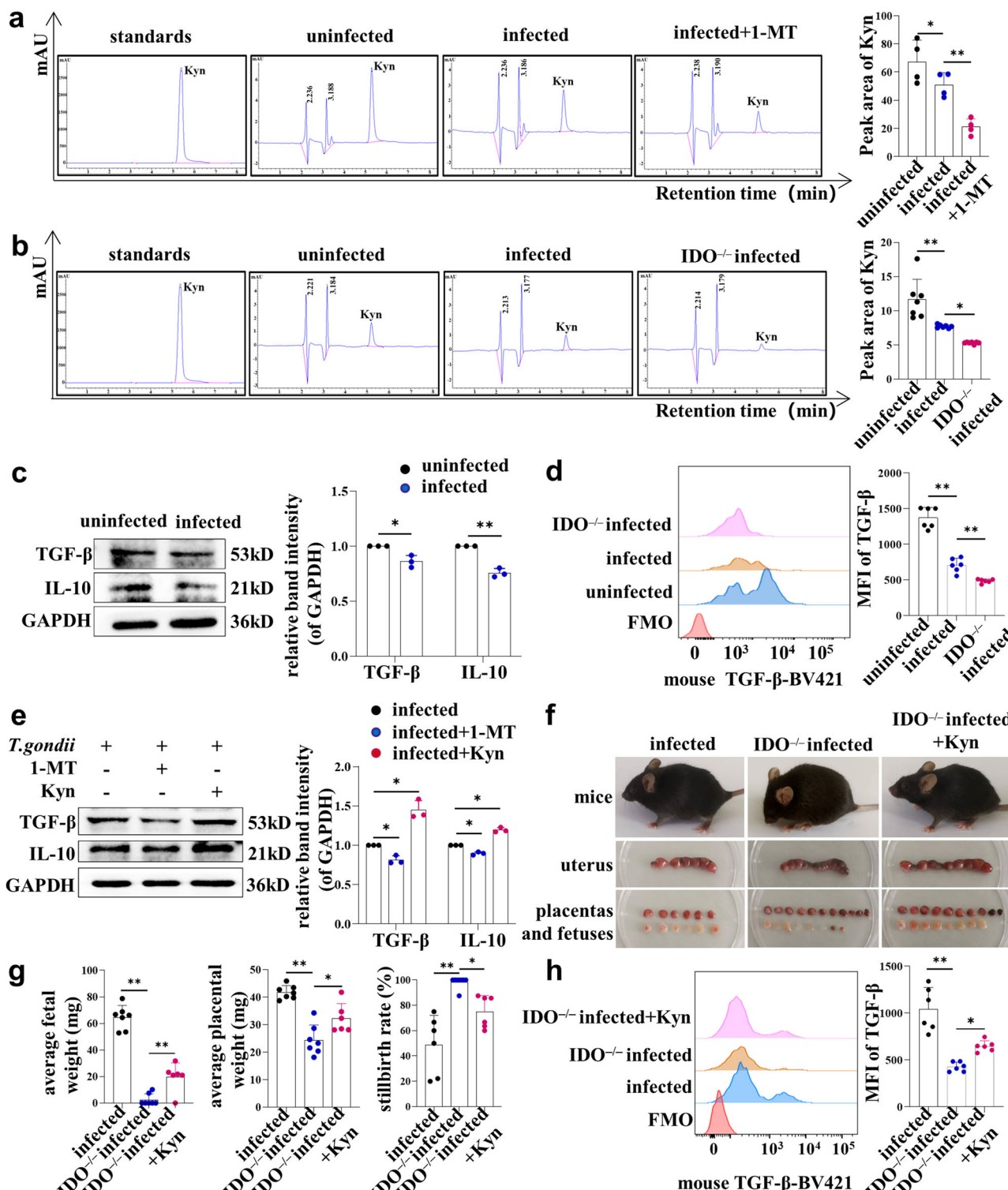

**Fig. 6 | The downregulation of IDO expression in dMDSCs after *T. gondii* infection could decrease expression levels of TGF-β and IL-10 in dNK cells. a** The peak area of Kyn in human dMDSCs of the uninfected, infected and infected with 1-MT groups were measured by HPLC (data represent the mean ± SD from four independent experiments by one-way ANOVA). **b** The peak area of Kyn in placental tissues in the uninfected, infected and IDO[−/−] infected groups was measured by HPLC (*n* = 7). **c** The expression changes of TGF-β and IL-10 in human dNK cells of co-culture system between uninfected group and infected group were detected by western blot (data represent the mean ± SD from three independent experiments by paired *t*-test). **d** The expressions of TGF-β in mouse dNK cells of uninfected, infected and IDO[−/−] infected mice were detected by flow cytometry (*n* = 6). **e** The

expressions of TGF-β and IL-10 in human dNK cells of the co-culture system in infected group, 1-MT-treated infected group and Kyn-treated infected group were detected by western blot (data represent the mean ± SD from three independent experiments by one-way ANOVA). **f** The uterus, developmental status of placentas and fetuses in the infected, IDO[−/−] infected and IDO[−/−] infected with Kyn treated groups. **g** The average weight of the placentas and fetuses and the abnormal fetuses rate in the three groups (*n* ≥ 6). **h** The expressions of TGF-β in dNK cells of infected, IDO[−/−] infected and Kyn-treated IDO[−/−] infected mice were detected by flow cytometry (*n* = 6). Data are presented as the mean ± SD, *\*P* < 0.05, *\*\*P* < 0.01, mice samples in each group assayed individually by one-way ANOVA. MFI: mean fluorescence intensity.

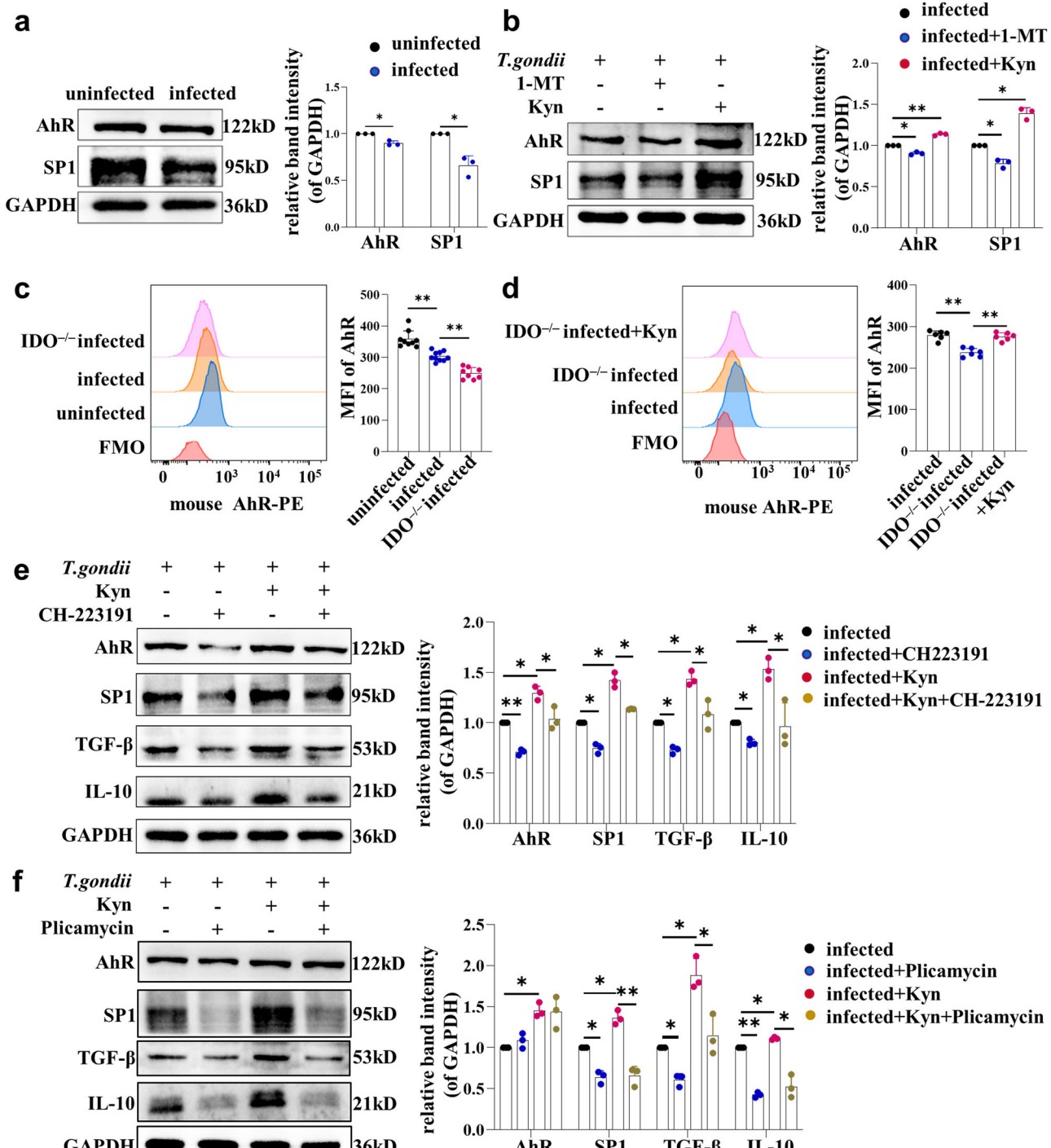

**Fig. 7 | The decreased of IDO expression in dMDSCs after *T. gondii* infection downregulated the expression of TGF-β and IL-10 in dNK cells through Kyn/AhR/SP1 pathway. a** The AhR and SP1 expressions in human dNK cells in the co-culture system were detected by western blot (data represent the mean ± SD from three independent experiments by paired *t*-test). **b** In the co-culture system, AhR and SP1 in human dNK cells were detected by western blot in infected group, 1-MT-treated infected group and Kyn-treated infected group (data represent the mean ± SD from three independent experiments by one-way ANOVA). **c** The AhR expressions in mouse dNK cells were analyzed by flow cytometry in uninfected ($n = 9$), infected ($n = 10$) and IDO$^{-/-}$ infected pregnant mice ($n = 8$). **d** The AhR expressions in mouse dNK cells were detected by flow cytometry in infected, IDO$^{-/-}$ infected and Kyn-treated IDO$^{-/-}$ infected pregnant mice ($n = 6$). The expressions of AhR, SP1, TGF-β and IL-10 in human dNK cells of the co-culture system were examined by western blot after treated with AhR inhibitor CH-223191 (**e**) or SP1 inhibitor Plicamycin (**f**), respectively (data represent the mean ± SD from three independent experiments by one-way ANOVA). Mice samples in each group assayed individually by one-way ANOVA. *$P < 0.05$, **$P < 0.01$, MFI: mean fluorescence intensity.

The results showed that the expression levels of TGF-β and IL-10 in dNK cells were significantly decreased after co-culturing with infected dMDSCs. Consistent with the results of in vitro, the expression level of TGF-β in infected mouse dNK cells was also significantly decreased compared to that of uninfected mice, and further decreased in infected IDO$^{-/-}$ mice. However, the immune mechanism by which the dysfunction of dNK cells induced by the decrease of IDO in dMDSCs during *T. gondii* infection also needs to be explored.

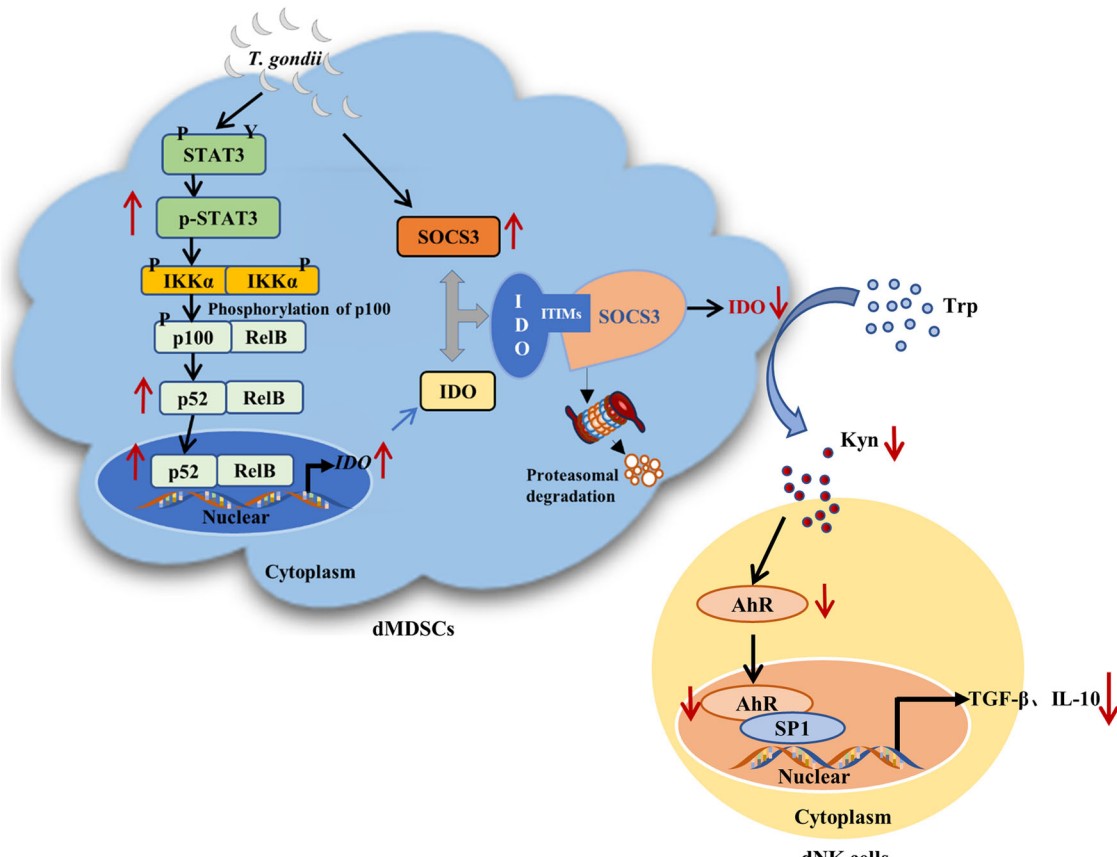

**Fig. 8 | A pattern diagram of the possible mechanism.** *T. gondii* infection upregulated the transcriptional levels of IDO in dMDSCs through STAT3/p52-RelB pathway. But IDO protein expression was decreased due to the degradation induced by the increase of SOCS3 by infection, which in turn the reduction of IDO metabolite kynurenine further downregulated TGF-β and IL-10 expression in dNK cells through Kyn/AhR/SP1 signaling pathway, ultimately leading to the dysfunction of dNK and the disorders of maternal-fetal tolerance.

IDO can degrade tryptophan to produce Kyn, the content of Kyn can be used as an indicator to assess IDO activity[11]. The Kyn contents in the supernatant of human dMDSCs and the placental supernatant of mice were both detected by HPLC. The results showed that the content of Kyn in infected human MDSCs was significantly lower than that in uninfected dMDSCs and further decreased after treatment with the IDO inhibitor 1-MT. Also, we found that the levels of Kyn in the placental supernatant of infected mice were obviously declined compared to those of uninfected mice, and further decreased in infected IDO$^{-/-}$ mice in vivo. These results demonstrated that the reduction of IDO in dMDSCs after *T. gondii* infection led to a decrease in Kyn production.

Kyn is an endogenous ligand of AhR and can promote the expression of AhR[11]. Hao et al. believed that the IDO/Kyn/AhR pathway is the core mechanism for establishing and maintaining maternal-fetal tolerance[54]. AhR is highly expressed in dNK cells and is involved in the normal development and growth of placental blood vessels[30]. Several studies found that the AhR/ARNT dimer recruited SP1 through combined with the zinc finger[39], and SP1 could bind to the promoters, thereby enhancing the expression of IL-10 and TGF-β[40–42]. Whether the downregulation of IDO in dMDSCs after *T. gondii* infection could regulate the expression of TGF-β and IL-10 in dNK cells through the Kyn/AhR/SP1 signaling pathway needs further investigation. The expression levels of AhR and SP1 in human dNK cells co-cultured with infected dMDSCs and AhR expression of mouse dNK cells were analyzed. We found that the expression levels of AhR and SP1 were significantly decreased in human dNK cells after co-culturing with infected dMDSCs. The expression levels of AhR in mouse dNK cells from infected mice were also significantly decreased, and they were further reduced in infected IDO$^{-/-}$ mice. These results suggested that the decrease of IDO in dMDSCs after *T. gondii* infection could downregulate the expression levels of AhR and SP1 in dNK cells. However, it was still unclear whether the dysfunction of dNK cells was regulated by the reduction of IDO in dMDSCs induced by *T. gondii* infection through the Kyn/AhR/SP1 signal pathway.

To investigate the effect of Kyn on the interaction between dMDSCs and dNK cells, dMDSCs were treated with an IDO inhibitor 1-MT or exogenous Kyn, respectively in the co-culture system. Interestingly, the results showed that the expression levels of AhR, SP1, TGF-β and IL-10 in dNK cells were all decreased or increased after treatment with 1-MT or exogenous Kyn. To our surprise, the adverse pregnancy outcomes of IDO$^{-/-}$ infected mice improved significantly after treatment with exogenous Kyn. There were less stillbirths and resorbed fetuses, and both the average weights of the placenta and fetal mice increased noticeably. Meanwhile, AhR and TGF-β expression levels in dNK cells from infected IDO$^{-/-}$ mice were significantly increased after treatment with exogenous Kyn. These above results suggested that the dysfunction of dNK cells were resulted from the decrease of IDO with *T. gondii* infection and the decline of Kyn content in MDSCs. To explore the molecular mechanism of the expression of TGF-β and IL-10 in human dNK cells was regulated by Kyn, AhR inhibitor CH-223191 or SP1 inhibitor Plicamycin was respectively added to the co-culture system. The results showed that the expressions of SP1, TGF-β and IL-10 were all decreased after treatment with CH-223191 or Plicamycin. But there was no obvious change in the expression of AhR after treatment with Plicamycin. Our results demonstrated that the downregulation of IDO in dMDSCs after *T. gondii* infection could regulate the expression of TGF-β and IL-10 in dNK cells through the Kyn/AhR/SP1 signaling pathway, resulting in the dysfunction of dNK cells, which may ultimately contribute to adverse pregnancy outcomes.

## Methods

### Ethics approval and consent to participate

The sample collection procedure for this study was approved by the Ethics Committee of Binzhou Medical University (approval number 2017-016-01). All ethical regulations relevant to human research participants were followed. Pregnant women who underwent voluntary abortions were enrolled in this study and signed informed consent forms. These participants were diagnosed with early normal gestation by a professional obstetrician and gynecologist, without any complications or ongoing medication, and with no existing infection by *T. gondii*. Animals' experiments were conducted according to the ethical standards of the Ethics Committee and Institutional Animal Experimental Ethics Committee of Binzhou Medical University (permit number 2017-009-09). We have complied with all relevant ethical regulations for animal use. All animal experiments were conducted under sodium pentobarbital anesthesia to minimize the animals' suffering.

### *T. gondii* RH strain

The *T. gondii* tachyzoites were stored in 85% ethylene glycol and frozen in liquid nitrogen. After post-resuscitation, the tachyzoites of *T. gondii* RH strain were harvested from the human foreskin fibroblast (HFF) cells cultured in high-glucose Dulbecco's modified Eagle's medium (DMEM) with 100 IU/ml penicillin/streptomycin (Procell, China) and 5% fetal bovine serum (FBS; Hyclone, USA). The cultures were collected after co-culturing HFF cells and toxoplasma tachyzoites for 72 h. The cells were removed by centrifugation at $433 \times g$ for 5 minutes (min), and the tachyzoites in the supernatant were purified by centrifugation at $2810 \times g$ for 7 min.

### Animals

Wide-type (WT) C57BL/6 mice (6–8-week-old females and 8–10-week-old males) were purchased from Pengyue Laboratory Animal Technology Co., Ltd. (Jinan, China). The IDO-deficient (IDO$^{-/-}$) mice were successfully bred by the Nanjing Model Animal Center. (Nanjing, China) with a C57BL/6 background. At the age of 4 weeks, the tail of each mouse was cut, and genomic DNA was extracted from the tail tissues using a DNA extraction kit (Generay, China). Polymerase chain reaction (PCR) synthesizes complementary DNA (cDNA). After an initial denaturation step (2 min at 94 °C), PCR was carried out with ten amplification cycles consisting of denaturation for 20 s at 94 °C, annealing for 15 s at 65 °C (−0.5 °C per cycle), and extension for 10 s at 68 °C. This was followed by 28 amplification cycles involving denaturation for 15 s at 94 °C, annealing for 15 s at 60 °C, and extension for 10 s at 72 °C, with a final extension step for 2 min at 72 °C. The PCR products were separated by electrophoresis in 1.8% agarose gels, and their sizes were estimated using Trans DNA Marker I (TransGen Biotech, China). To visualize DNA, the gels were stained with GelStain (TransGen Biotech, China). The primers for PCR amplification were as follows: TGGAGCTGCCCGACGC; TACCTTCCGAGCCCAGACAC; CTTGGG TGGAGAGGCTATTC; AGGTGAGATGACAGGAGATC. The expected PCR product sizes were 280 bp (mutant), 280 and 427 bp (heterozygote), and 427 bp (WT). The obtained homozygous IDO$^{-/-}$ mice were bred and used to establish animal models. All mice were raised in a specific pathogen free (SPF) animal facility at Binzhou Medical University, maintained at a temperature of 24 °C ± 2 °C with 55% ± 5% humidity. They were provided with ample sterilized water and food (Keao Yanli Feed Co., Ltd., China).

### Pregnant mice models

In this study, 6–8-week-old females and 8–10-week-old males were caged together at a female-to-male ratio of 2:1. On the following day, the females with a vaginal plug were designated as gestational day (Gd) 0 of pregnancy. The infected mice were injected intraperitoneally (*i.p.*) with 200 tachyzoites resuspended in 200 μl of phosphate buffered saline (PBS) on Gd 7, while the uninfected mice received with the same volume of PBS injected into the abdominal cavity. The animal models of WT and IDO$^{-/-}$ pregnant mice infected with *T. gondii* infection were established. The fetal and maternal tissues were harvested on Gd 13. The pregnancy outcomes were observed,

the weights of the placenta and fetus were measured, and the tissues of the uterus and placenta were collected for flow cytometry detection.

For treatment with the STAT3 inhibitor, cucurbitacin I (JSI-124; MCE, China), the pregnant mice were *i.p.* injected with 200 tachyzoites on Gd 7 and were treated *i.p.* with 1 mg/kg of JSI-124 three times a week (Gd 7, Gd 9, and Gd 11). For treatment with exogenous kynurenine (Kyn; MCE, China), 2 mg/ml of Kyn was added to the feeding water of IDO$^{-/-}$ pregnant mice from Gd 7 to Gd 13.

### Sample cell preparation

On Gd 13, the uterus and placental tissue were carefully separated and washed three times with cold PBS. The tissues were cut into 1–3 mm fragments using ophthalmic scissors. Then, the fragments were suspended in RPMI 1640 medium, 1 mg/ml of collagenase IV (Biofroxx, Germany) and 0.2 mg/ml of DNase I (Sigma-Aldrich, St. Louis, USA) were added. The samples were digested in a biochemical incubator at 37 °C for 40 min. The digested tissues were filtered through 48 μm sterile nets and collected in 50 ml centrifuge tubes. The cells were removed by centrifugation at $700 \times g$ for 10 min, the supernatant was discarded, and the precipitate was resuspended in cold PBS. The mononuclear cells were collected from a white membrane layer in the mouse lymphocyte isolation medium (TBD Science, China) after Ficoll density gradient centrifugation. Finally, the cells were collected, resuspended in cold PBS, and used for subsequent flow cytometry analysis.

### Human samples

The human decidual tissues from healthy pregnant women at 6–8-weeks of gestation were collected. The participants underwent voluntary abortions at the abortion clinic of Yantai Affiliated Hospital, Binzhou Medical University, the Yantai Zhifu District Material and Child Health Hospital.

### Isolation and purification of human dMDSCs and dNK cells

The decidual tissues were washed 4–5 times with cold PBS to remove the blood and then cut into small fragments measuring 1–3 mm using ophthalmic scissors. Then, the fragments were suspended in RPMI 1640 medium, 1 mg/ml of collagenase IV (Biofroxx, Germany) and 0.2 mg/ml of DNase I (Sigma-Aldrich, St. Louis, USA) were added to digest the tissues in a biochemical incubator at 37 °C for 50–60 min. The digested tissues were filtered through 48 μm sterile nets and collected in 50 ml centrifuge tubes. The cells were removed by centrifugation at $700 \times g$ for 10 min, the supernatant was discarded, and the precipitate was resuspended in cold PBS. The mononuclear cells were collected from the white membrane layer using human lymphocyte isolation medium (TBD Science, China) after Ficoll density gradient centrifugation. Finally, the cells were collected, resuspended in cold PBS, and used for subsequent flow cytometry analysis. According to the protocol, the dMDSCs were purified using a human HLA-DR-negative and CD33-positive selection kit (Stem Cell Technologies, Canada). The dNK cells were purified using human CD3-negative and CD56-positive selection kits (Stem Cell Technologies, Canada) according to the manufacturer's instructions.

### Human decidual cell culture and treatment

The decidual mononuclear cells and dMDSCs were divided equally into uninfected and infected groups (at a 1:5 ratio of *T. gondii*: cells). All cells were cultured in RPMI 1640 medium supplemented with 10% FBS, 100 IU/ml penicillin/streptomycin for 24 h at 37 °C in a humidified 5% $CO_2$ incubator. The decidual mononuclear cells were prepared for flow cytometry staining and analysis. The purified dMDSCs were used for quantitative real-time PCR (qPCR), Co-immunoprecipitation (CO-IP) and western blot assays.

The purified human dMDSCs were added to the upper chamber (0.4 μm pore size; Corning, America, cat:3450) of a transwell six-well plate with 2 ml of medium. Meanwhile, the purified human dNK cells were added to a transwell subplate chamber with a 2 ml medium system (at a 1:2 ratio of dMDSCs: dNK). The upper compartment dMDSCs of the infected groups were infected with *T gondii* (*T. gondii*: cells, 1:5) and cultured for

approximately 24 h. Subsequently, cells from the upper and lower compartments were collected for western blot assay.

## Flow cytometry

For experiments involving measuring IDO and p-STAT3 in mouse dMDSCs, AhR and TGF-β in mouse dNK cells, the following fluorochrome-conjugated monoclonal antibodies (mAbs) were used: PERCP-CY5.5-CD11b (Biolegend, USA, clone: M1/70), FITC-Gr1 (Biolegend, USA, clone: RB6-8C5), eFluor 660-IDO (Invitrogen, USA, clone: mIDO-48), APC- p-STAT3 (Tyr705) (Invitrogen, USA, clone: LUVNKLA), PERCP-CY5.5-CD3e (Invitrogen, USA, clone: 145-2C11), APC-CD122 (Invitrogen, USA, clone: TM-beta1), FITC-CD49b (Biolegend, USA, clone: DX5), PE-AhR (Invitrogen, USA, clone: 4MEJJ) and BV421-TGF-β (Biolegend, USA. clone: TW7-16B4). In vitro, human-specific mAbs were used: PE-CD33 (Biolegend, USA, clone: WM53), FITC-HLA-DR (Invitrogen, USA, clone: LN3), APC-IDO (Invitrogen, USA, clone: eyedio), APC-p-STAT3 (Tyr705) (Invitrogen, USA, clone: LUVNKLA) and BV421-p-STAT3 (Tyr705) (Biolegend, USA, clone: 13A3-1). The prepared human and mice decidual mononuclear cell suspensions were stained with surface markers. After surface marker staining, the cells were fixed and permeabilized using CytoFix/Perm solution (Foxp3/Transcription Factor Staining Buffer Set, eBioscience) and stained with p-STAT3, IDO, AhR and TGF-β antibodies following the manufacturer's instructions. A FACSCanto II flow cytometer (BD Bioscience) was used for all flow cytometry data acquisition, and the data were analyzed using FlowJo analysis software (FlowJo, USA). The gating strategies are provided in Supplementary Fig. 1.

## Quantitative real-time PCR (qPCR)

The total RNA was extracted from human decidual dMDSCs using TRIzol reagent (Invitrogen). The RNA was transcribed into cDNA using a SuperRT cDNA Synthesis Kit (CWBIO) following the manufacturer's instructions. Quantitative real-time PCR (qPCR) was performed using an UltraSYBR One-Step RT-qPCR Kit (CWBIO) in a Bio-Rad iQ5 multicolor RT-PCR system. The GAPDH messenger RNA (mRNA) expression was detected in each experimental sample as an endogenous control. The primers used for qPCR were as follows: GAPDH (F: 5'- GATTTGGTCGTATTGGGCGC -3' and R: 5'- TTCCCGTTCTCAGCCTTGAC -3'), IDO (F: 5'- GCCAGCTTCGAGAAAGAGTTG -3' and R: 5'- ATCCCAGAACTA-GACGTGCAA -3'), p52 (F: 5'- CTACTCGACTACGGCGTCAC -3' and R: 5'- GTTGGTGAGGTTGACAACGC -3'), RelB (F: 5'- CATGGCATC-GAGAGCAAACG -3' and R: 5'- GACACGGTGCCAGAGAAGAA -3'), SOCS3 (F: 5'- CATCTCTGTCGGAAGACCGTCA -3' and R: 5'- GCATCGTACTGGTCCAGGAACT -3').

## Isolation of nuclear proteins

Nuclear extracts of dMDSCs were prepared using Nuclearand Cytoplasmic Protein Extraction kit according to the manufacturer's instructions (Beyotime Biotechnology, China). The experimental procedures were as followed:(1) the dMDSCs between groups were washed with ice-cold PBS twice, detached and collected; (2) after centrifugation at $700 \times g$ for 10 min at 4 °C, the supernatant was discarded and the cells were fully resuspended in 200 µl of cytoplasmic protein extraction reagent (CPER) A;(3) the cell suspensions were dispersed by vortexing for 5 s and incubated on ice for 10–15 min; (4) 10 µl of CPER B was added to each tube followed by vigorous vortexing for 5 s and incubated on ice for 1 min; (5) the cell lysates were centrifuged at $14,000 \times g$ for 10 min at 4 °C and the supernatant was cytoplasmic proteins; (6) the insoluble cell pellets were resuspended in 50 µl nuclear protein extraction reagent and successively vortexed for 20 s every 5 min for a total of 30 min on ice;(7) the cells were centrifuged at $14,000 \times g$ for 10 min at 4 °C, the supernatant containing the extracted nuclear proteins was mixed with 1× loading buffer and boiled for 10 min at 100 °C for the subsequent western blot assay.

## Western blot

The cells were lysed using RIPA lysis buffer (Beyotime Biotechnology, China) supplemented with 1× phenylmethylsulfonyl fluoride (PMSF;

Beyotime Biotechnology, China) on ice for 45–60 min and then centrifuged at $12,000 \times g$ for 20 min at 4 °C. The supernatant was preserved, and the protein concentrations of the extracts were measured using the bicinchoninic acid assay (Beyotime Biotechnology). A whole-cell lysate (WCL) supernatant was prepared for the input sample with 1× loading buffer, boiled for 10 min at 100 °C. Equal amounts of proteins were loaded onto 10% or 12% SDS-PAGE gels and transferred to polyvinylidene fluoride (PVDF) membranes. After the membranes were blocked at room temperature for 2–3 h in 5% nonfat dry milk in TBS-T buffer, they were incubated overnight with primary antibodies at 4 °C. The membranes were washed 5 times for 5 min with TBS-T buffer and then incubated for 2 h at room temperature with the appropriate HRP-conjugated secondary antibody. Subsequently, the membranes were washed 5 times for 10 min with TBS-T buffer. The immunoreactive bands were visualized using an enhanced chemiluminescence (ECL) detection kit (Yeasen) and analyzed with the Bio-Red ChemiDoc XRS⁺ System. The primary antibodies used included STAT3 (Proteintech, China, 10253), p-STAT3 (CST, UK, 9145S), IDO (Abcam, UK, ab76157), IKKα (Wanleibio, China, WL00053), p- IKKα (ABclonal, China, AP0506), p-p100 (ABclonal, China, AP1367), p52 (Proteintech, China, 15503), RelB (Proteintech, China, 66947), SOCS3 (Wanleibio, China, WL01364), GAPDH (Proteintech, China, 10494), LaminB (Wanleibio, China, WL01775), AhR (Wanleibio, China, WL02657), SP1 (Proteintech, China, 21962), TGF-β (Bioss, China, bs-0086R) and IL-10 (Wanleibio, China, WL03088).

## Co-immunoprecipitation (CO-IP) assays

The total protein was lysed in RIPA lysis buffer and 1× PMSF on ice for 40–60 min and then centrifuged at $14,000 \times g$ at 4 °C for 15–20 min. The supernatant was collected and divided into WCL and immunoprecipitation (IP) groups. The supernatant of the IP group was incubated with anti-IDO (Abcam, UK, ab211017) or IgG as a negative control at 4 °C overnight. The antibody concentration was 5 µg/ml and precipitated with protein A/G-agarose beads overnight at 4 °C. The beads were washed three times with a washing buffer using a magnetic separator. The precipitated proteins were denatured in 1× loading buffer and analyzed by western blotting.

## High performance liquid chromatography (HPLC)

The human dMDSCs were collected from each group, centrifuged at $12,000 \times g$ for 20 min, and the supernatants were carefully absorbed. Placenta (100 µg) from pregnant mice in each group was collected and placed into a 1.5 ml EP tube with 1 ml of PBS, the placental tissues were crushed using a homogenizing machine, then centrifuged at $12,000 \times g$ for 20 min, and the supernatants were carefully collected. Around 1 mM of Kyn standard solution was prepared. The chromatographic separation was performed at 25 °C using an Eclipse XDB-C18 (4.6 mm × 150 mm, 5 µm) and a mobile phase consisting of 10 mmol/L sodium acetate-acetic acid buffer (pH 4.53) and acetonitrile with a volume ratio of 93:7 at a flow rate of 1.0 ml/min. The detection was performed using a spectrophotometric detector at wavelengths of 365 nm for Kyn over a period of 0–15 min. A 50 µL aliquot of the supernatant was injected into the HPLC system to acquire the peak area. After observing the retention time, the Kyn content of each experimental tube was detected, and the peak area of Kyn of each group was recorded.

## Inhibition of pathway assay

To investigate if IDO expression was regulated by the STAT3/p52-RelB pathway after *T. gondii* infection, the p-STAT3 inhibitor (stattic; MCE, China) and p52-RelB nuclear translocation inhibitor (SN52; MCE, China) were separately introduced to human dMDSCs. To investigate the role of the Kyn/AhR/SP1 pathway, the co-cultured cells were divided into five groups: infected, IDO inhibitor (1-methyl-tryptophan, 1-MT; Sigma-Aldrich, UK)-treated infected, exogenous kynurenine (Kyn; MCE, China)-treated infected, AhR inhibitor (CH-223191; MCE, China)-treated infected, and SP1 inhibitor (Plicamycin; MCE, China)-treated infected groups. All samples were cultured in RPMI 1640 medium supplemented with 10% FBS and 100 IU/ml penicillin/streptomycin for 24 h at 37 °C in a humidified 5% $CO_2$ incubator.

## Statistics and reproducibility

All data in this study were obtained from at least 3 independent replicate experiments, and are presented as the mean ± standard deviation (SD). Data points in Figures represent biological replicates. The statistical analyses were performed using the GraphPad Prism 9 statistical software package. Two-tailed Student's $t$ test or one-way ANOVA was used for the data. The statistical significance in the figures is shown as $^*P < 0.05$ or $^{**}P < 0.01$.

## Reporting summary

Further information on research design is available in the Nature Portfolio Reporting Summary linked to this article.

## Data availability

All data generated in this study are presented within this published article. The source data can be found in Supplementary Data 1 and all uncropped blots can be found in Supplementary Fig. 2. The original experimental raw data can be obtained from the authors upon request.

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

## Acknowledgements

This work was supported by funds from the National Natural Science Foundation of China (NO. 81871680, NO. 81672049), Taishan Scholar Foundation of Shandong province (NO. ts201712066) and Natural Science Foundation of Shandong province (NO. ZR2021QH039).

## Author contributions

Y.W., X.Y.Z., Z.D.L. and X.M.H. designed the experiments; Y.Z.J., X.B.L. and X.Y.X. contributed to sample collection; W.X.W., H.X.Z. and Y.S.R. analyzed the data; Y.W., X.Y.Z., Z.D.L. and X.M.H. wrote the manuscript; Y.W., X.Y.Z. and X.M.H. edited the manuscript. X.M.H. as the corresponding author. All authors read and approved the final manuscript.

## Competing interests

The authors declare no competing interests.
