## [Peer Review File · Communications Biology]

Reviewers' comments:

Reviewer #1 (Remarks to the Author):

This is a really interesting study looking at IDO in pregnancy and infection with *T. gondii*.

However there are still some problems in the study that need to be addressed before the paper would be acceptable for publication.

Major Points:

1. Why do the authors use CD49b as a marker for NK cells in the decidua? Why did the authors not use NK1.1 for the staining. In the supplemental figure looking at the gating, the authors should also maybe also be more conservative with gating the T cells since CD49b+ T cells may still exist and in general have low CD3 expression which might contaminate their current gate. Also the CD49b staining looks very weak, the authors should maybe reconsider using another fluorophore or as stated above use another NK cell marker?

Also the authors gate on both a high and low population of CD33 expressing cells from the human decidua. I would have thought that the high expressing cells were the MDSC? Are there differences in expression of IDO etc if the authors look at the high and low populations?

2. The authors point out that the IDO mice suffer more from *T. gondii* infection and bring up a point about their mental status. What exactly was their mental status (depression)? Is it cannot be measured I suggest that the author remove this from the text. In addition the authors do not report on parasite load in the different strains of mice. Do the IDO mice have a higher parasite load which might explain also the severity of the responses in these mice?

3. In a number of the figures where the authors discuss the protein levels, the authors write, relative levels. Could the authors not set these numbers to 1 for the uninfected and then show the relative level from the infected cells.

4 In the mixed cells experiments with MDSC and NK cells, the authors do not outline the size of the pores in the transwell. I assume the size would prevent cells from crossing but would it stop tachyzoites from crossing should they induce cell death in the MDSCs. The group of Barragan has shown that dendritic cells infected with *T. gondii* become hypermotile and NK cells can be infected with *T. gondii* which alters their function. Are the authors sure that there are no infected NK cells?

It would also be good if the authors could potentially show that there are infected MDSC in the decidua and that the effect that they are seeing is not a systemic effect due to infection.

5. I am also concerned with the statistics as the authors do not say how many mice are used or how many times that they have performed the human experiments. Since they are using bar charts it is difficult to see the points. For me I would increase the number of experiments to improve the statistics if

the experiments are performed less than 6 times for the human experiments or 10 times for the mouse experiments.

Minor problems:

1. The language in some parts was difficult to read and should be corrected. I had started making points of which lines etc but then became too many, so might be easier on revision for the authors to correct then

2. I liked the model that was presented but the authors could add more description in the Figure Legend

Reviewer #2 (Remarks to the Author):

This work is dedicated to the study of the influence of decidual NKs on the expression of Indolamine-2,3-dioxygenase (IDO) by myeloid suppressor cells (MDSCs) in a context of *Toxoplasma gondii* infection during pregnancy. The authors pose several questions which they answer successively through experiments using the mouse animal model and human ex-vivo co-cultures. Thus, the influence of IDO on the outcomes of pregnancy during *T. gondii* infection was evaluated using a murine model of pregnancy (C57Bl6 IDO deficient). This model made it possible to monitor the consequences of *T. gondii* infection on pregnancy development parameters such as fetal weight, placental weight, and stillbirth rate in the absence/presence of IDO. Ex vivo experimental models using primary co-cultures of NKs and MDSCs derived from human decidua tissues are used, and fundamental and standard methodologies for characterizing these cell populations (flow cytometry) allowed IDO expression to be assessed. Studies of the mechanisms involving nuclear transcription factor activation pathways (STAT3/p52-RelB), IDO/kynurenine/ Ahr pathway associated with the analysis of IDO expression were carried out using western blot and qPCR. It was also studied how the alteration in IDO expression found in dMDSCs could regulate the production of cytokines such as IL10, TGF beta by dNKs through the Kynurenine/AhR/SP1 pathway with the use of inhibitors. The production of kynurenine was evaluated by HPLC.

In recent years, this team of researchers has focused on studying the changes in various cell populations such as NKs, regulatory T cells, dendritic cells and macrophages at the level of the fetal-maternal interface that occur during infection with *Toxoplasma gondii*, a parasite that is still underestimated, but which influences and may have serious consequences for pregnancy, for the development of the fetus and, in the case of congenital infection, the individual may develop later in life psychological alterations and, in the case of immunosuppression, reactivation of the infection and development of severe encephalitis which may even lead to death.

Recently, a negative regulation of Arg-1 and IL-10 by decidual MDSCs through the Fyn-STAT3-C/EBP β signaling pathway has been described, leading to a decrease in the immunosuppressive function of dMDSCs and, therefore, the increase of adverse effects in pregnancy. Other studies have demonstrated the role of these cells through the inhibitory molecule leukocyte immunoglobulin-like receptor B4

(LILRB4), which regulates the expression of Arg-1 and IL-10 through the SHP-2/STAT6 pathway, resulting in the dysfunction of dMDCs, which could contribute to adverse outcomes during pregnancy due to *T. gondii* infection (<https://doi.org/10.1186/s13071-023-05856-4>). Recently, it was also described that IDO activation by mature cDC1s promoted a tolerogenic profile of inflammatory cDC2s through metabolic communication via their production of the tryptophan metabolite L-kynurenine. This fact is not mentioned in this document (doi: 10.1016/j.immuni.2022.05.013).

This work is novel in that it analyses in greater detail the expression of IDO, which together with Arg1 and IL10 contributes to the immunosuppressive properties of dMDCs during pregnancy, and its influence on dNKs, which are one of the main populations at the fetal-maternal interface that determine the proper course of pregnancy. It seems that this will be an important piece of work for the scientific community and the development of new knowledge in this area, which is essential for understanding the major players during pregnancy. The results could also be used to understand the cell populations status at the fetus-maternal interface that are certainly involved with other pathogens that establish infection and the development of pathology associated with interference in the biological processes during pregnancy. However, the phenotypic characterization of dMDCs is based exclusively on the expression of HLADR and CD33 and is therefore very preliminary. This analysis will have to be carried out more thoroughly in order to draw clearer conclusions about the role of dMDCs in the experimental conditions tested (<https://doi.org/10.1002/cpz1.561>). This document lacks information about some items included in the reporting summary of nature portfolio, namely for statistical analysis, some of the conclusions and statements are not clear such as "line 27-28: ...but obviously decreased..."); the co-immunoprecipitation procedures are not clearly explained (line 306-307: what does WCL mean?); some phrases are not understood (Examples: line 66-67; line 83-84; line 93-94: "all immune cells interact with each other..."; repetition of information: line 96-100 repeating information from lines 89-91; is not referenced the clones used of monoclonal antibodies for the experimental procedures. Is not understood the parameters evaluated for the mice mental status; is referred about the expression of nuclear expression of p52 and RelB by western blot, but is not described the organelle isolation procedure in the document; is referred that recently a novel decidual (line 600) MDSCs referring to references from 2017; Overall, this document has some strong points, in that the study of the interconnection between dMDCs and dNKs should play determinant role during *T. gondii* infection, and the adjacent inflammatory response at the fetal-maternal interface and how this response influences metabolism and the production of metabolites such as Kynurenine which is also important to pregnancy and influence the development of the fetus. At present, there is still controversy in the characterization of dMDCs, so precautions should be taken in the conclusions taken from the data obtained. The phenotypical characterization used in this study include different profiles of dMDCs that shared the same protein profile expression with other co-existing populations at the fetal-maternal interface such as dDCs.

Reviewer #3 (Remarks to the Author):

The present work considers the effect of *Toxoplasma gondii* on MDSCs cells and the collateral effect towards dNKs. It is an innovative topic, important for the understanding of dNK immunomodulation in the presence of infected MDSCs cells. I have some points of view that are not entirely clear to me.

- 1.- In material and methods it mentions that they are marked "CD33-positive" to purify dMDSCs. Can you explain if this mark is sufficient to purify dMDSCs since this CD is expressed by myeloid stem cells, myeloblasts and monoblasts, monocytes/macrophages, granulocyte precursors and mast cells, and classical neutrophils?
- 2.-Likewise, you mention that dNKs are marked with "CD3-negative and a CD56-positive selection." Can you explain if this mark is sufficient to purify dNK since they are markers that cross with other cells?
- 3.-Most of the graphs show very large deviations but are statistically significant. Furthermore, they do not mention the sample size used, both in mice and in human placentas. Therefore, I suggest checking if the statistical method used is the most appropriate for this work.
- 4.- Results section 1st mentions "including extremely poor mental status" This statement is unclear and very subjective, and there is no context of how it was evaluated.
- 5.- Figure 4b presents different banding patterns with a stat3 inhibitor. My question is why it was not completely inhibited if it is an inhibitor of stat 3 phosphorylation?
- 6.- Figure 6b They evaluate the effect of IDO -/- on kyn, it is observed that the uninfected kyn is lower in mouse placentas than in human placentas, my question is why this effect if in figure 2c, d; it is observed that there is more IDO in mouse placentas than in those isolated from humans.
- 7.-In the Supplementary Fig. 1 section, you can see 2 populations, how did you consider that the two populations are dMSCs?
- 8.-Finally they demonstrated the increase to mRNA of IDO (IDO protein decrease) and SOCS3 in cells infected with toxoplasma gondii, but it is not discussed how the parasite could interact with the cell to obtain this.

Reviewer 1

This is a really interesting study looking at IDO in pregnancy and infection with *T. gondii*. However there are still some problems in the study that need to be addressed before the paper would be acceptable for publication.

Major Points:

Comment 1. Why do the authors use CD49b as a marker for NK cells in the decidua? Why did the authors not use NK1.1 for the staining. In the supplemental figure looking at the gating, the authors should also maybe also be more conservative with gating the T cells since CD49b⁺ T cells may still exist and in general have low CD3 expression which might contaminate their current gate. Also the CD49b staining looks very weak, the authors should maybe reconsider using another fluorophore or as stated above use another NK cell marker?

Response 1-1: Thanks for your comments.

It was known that uNK cells was first defined as CD3⁻CD122⁺ cells, other markers like Dolichus biflores agglutinin (DBA), NK1.1, and CD49b were used to distinguish into different subsets. Study has shown that the unusual NK1.1⁻CD49b⁻ uNK cell subset is dynamic and increases through pregnancy, peaking at mid-gestation (~70%) (<https://doi.org/10.4049/jimmunol.181.9.6140>), which was in line with the time we detected uNK cells in animal experiment. In addition, a another study also showed that CD49b⁺ uNK subsets were only a minor subset during normal pregnancy in the mouse, which represent cytotoxic function, and the CD49b⁻ NK cells was a majority of uNK cells which mainly exert immunotolerant function (<https://doi.org/10.1038/cmi.2014.44>). In our study, we aimed to explore the effect of IDO expressed in dMDSCs in the tolerant function of dNK cells during *T. gondii* infection. So, we ultimately used CD3⁻CD122⁺CD49b⁻ cells to gate decidual NK cells. In our study, CD3⁻CD122⁺ was used to gate total decidual NK cells. As you suggested CD49b⁺ T cells may still exist and in general have low CD3 expression, but CD3⁻CD122⁺ dNK cells were not included CD49b⁺ T cells. Then CD3⁻CD122⁺CD49b⁻ or CD3⁻CD122⁺ CD49b⁺ cells were usually used to distinguish the decidual NK subset

cells with immunotolerant function or cytotoxic function. In our study, we only gate CD3⁻CD122⁺CD49b⁻ dNK cells as total dNK cells to be analyzed, because CD3⁻CD122⁺CD49b⁺ cells were very little contained in decidual NK cells. Also, we re-analyzed the gate of dNK cells as shown below, CD3⁻CD122⁺CD49b⁻ cells were approximately 90.2% while CD3⁻CD122⁺CD49b⁺ cells were only 7.6%.

Also the authors gate on both a high and low population of CD33 expressing cells from the human decidua. I would have thought that the high expressing cells were the MDSC? Are there differences in expression of IDO etc if the authors look at the high and low

populations?

Response 1-2: MDSC is a heterogeneous group of immature myeloid cells, which are the precursors of dendritic cells, macrophages and granulocytes (<https://doi.org/10.1016/j.immuni.2021.04.004>). In general, the phenotype of MDSC was CD33⁺ HLADR⁻/^{dim} cells (<https://doi.org/10.1016/j.jri.2021.103316>). Abundance research also reported that CD33 high or dim cells were both represent the subsets of MDSC, CD33^{high} expressed MDSC represent mononuclear phagocyte-like MDSCs (M-MDSCs), CD33^{dim} subsets of MDSC belong to polymorphonuclear-like MDSCs (PMN-MDSCs) (<https://doi.org/10.3389/fimmu.2020.01345>; <https://doi.org/10.1038/ncomms12150>; <https://doi.org/10.1158/1078-0432.CCR-17-3726>; <https://doi.org/10.1126/scitranslmed.aae0482>). According to your question, we analyzed the differences in expression of IDO in high and low populations. Our results showed IDO were both highly expressed in the two subsets of dMDSCs and were all downregulated after *T. gondii* infection (as below). So, we analyzed the expression of IDO in combined CD33 positive dMDSC and the effect of IDO in dMDSC function. Thanks for your valuable comments, we will further explore the differences that IDO in CD33 high- or low-expressed dMDSCs in further study.

Comment 2. The authors point out that the IDO mice suffer more from *T. gondii* infection and bring up a point about their mental status. What exactly was their mental status (depression)? Is it not measurable? I suggest that the author remove this from the text. In addition, the authors do not report on parasite load in the different strains of mice. Do the IDO mice have a higher parasite load which might explain also the severity

of the responses in these mice?

Response 2:

(1) Thank you very much for your constructive suggestions, the mental status of the IDO mice couldn't be measured, we have removed the description in manuscript.

(2) In our previous study, we analyzed the parasite load of mice infected by *T. gondii* of WT and IDO^{-/-} mice by Real-time PCR. There was no difference. So, we think that the severity of abnormal pregnancy outcomes in IDO^{-/-} mice was due to the dysfunction of decidual immune cell and maternal-fetal tolerance.

Comment 3. In a number of the figures where the authors discuss the protein levels, the authors write, relative levels. Could the authors not set these numbers to 1 for the uninfected and then show the relative level from the infected cells.

Response 3: Thanks for your suggestion. The relative levels of protein we mentioned in data refer to the ratio of each protein to its corresponding internal reference (GAPDH). We have revised these into "Protein/GAPDH" in all figures.

Comment 4. In the mixed cells experiments with MDSC and NK cells, the authors do not outline the size of the pores in the transwell. I assume the size would prevent cells from crossing but would it stop tachyzoites from crossing should they induce cell death in the MDSCs. The group of Barragan has shown that dendritic cells infected with *T. gondii* become hypermotile and NK cells can be infected with *T. gondii* which alters their function. Are the authors sure that there are no infected NK cells?

It would also be good if the authors could potentially show that there are infected MDSC in the decidua and that the effect that they are seeing is not a systemic effect due to infection.

Response 4: Thanks for your suggestion, we have added the detailed information about the size of the pores in the transwell in lines 229-230. The aperture of this transwell plate was 0.4 μm, while the size of *T. gondii* size was (2-4) × (4-7) μm, so *T. gondii* couldn't transmit the well from the upper layer into the down layer to infect dNK cells. The results of Barragan group showed that dNK cells were infected by transmitted *T. gondii* was due to cell-cell contact directly. Hence, in our study, transwell assay was

used to prevent *T. gondii* from entering the lower compartment infecting dNK cells. As we know, *T. gondii* can be vertical transmitted from mother to fetus. In our present study the status of *T. gondii* infected MDSC of decidua was not analyzed. We think the decidual MDSC could be infected by *T. gondii*. Thanks for your suggestion, we will explore the infected status of decidual MDSC and that the systemic effect due to infection in the next study.

Comment 5. I am also concerned with the statistics as the authors do not say how many mice are used or how many times that they have performed the human experiments. Since they are using bar charts it is difficult to see the points. For me I would increase the number of experiments to improve the statistics if the experiments are performed less than 6 times for the human experiments or 10 times for the mouse experiments.

Response 5: Thanks for your kind suggestion. We have added the detailed information about the number of mice or human sample in each figure legend and changed the bar charts type to make it easier to see the points.

Minor Points:

Comment 1. The language in some parts was difficult to read and should be corrected. I had started making points of which lines etc but then became too many, so might be easier on revision for the authors to correct then

Response 1: Thanks for your suggestion, we have revised the language carefully and corrected the improper points in manuscript.

Comment 2. I liked the model that was presented but the authors could add more description in the Figure Legend

Response 2: Thanks for your suggestion, we have added detailed description about the model in Figure legend 8 in lines 607-613.

Reviewer 2

This work is dedicated to the study of the influence of decidual NKs on the expression of Indolamine-2,3- dioxygenase (IDO) by myeloid suppressor cells (MDSCs) in a

context of *Toxoplasma gondii* infection during pregnancy. The authors pose several questions which they answer successively through experiments using the mouse animal model and human ex-vivo co-cultures. Thus, the influence of IDO on the outcomes of pregnancy during *T. gondii* infection was evaluated using a murine model of pregnancy (C57Bl6 IDO deficient). This model made it possible to monitor the consequences of *T. gondii* infection on pregnancy development parameters such as fetal weight, placental weight, and stillbirth rate in the absence/presence of IDO. Ex vivo experimental models using primary co-cultures of NKs and MDSCs derived from human decidua tissues are used, and fundamental and standard methodologies for characterizing these cell populations (flow cytometry) allowed IDO expression to be assessed. Studies of the mechanisms involving nuclear transcription factor activation pathways (STAT3/p52-RelB), IDO/kynurenine/ Ahr pathway associated with the analysis of IDO expression were carried out using western blot and qPCR. It was also studied how the alteration in IDO expression found in dMDSCs could regulate the production of cytokines such as IL10, TGF beta by dNKs through the Kynurenine/AhR/SP1 pathway with the use of inhibitors. The production of kynurenine was evaluated by HPLC. In recent years, this team of researchers has focused on studying the changes in various cell populations such as NKs, regulatory T cells, dendritic cells and macrophages at the level of the fetal-maternal interface that occur during infection with *Toxoplasma gondii*, a parasite that is still underestimated, but which influences and may have serious consequences for pregnancy, for the development of the fetus and, in the case of congenital infection, the individual may develop later in life psychological alterations and, in the case of immunosuppression, reactivation of the infection and development of severe encephalitis which may even lead to death.

Recently, a negative regulation of Arg-1 and IL-10 by decidual MDSCs through the Fyn-STAT3-C/EBP β signaling pathway has been described, leading to a decrease in the immunosuppressive function of dMDSCs and, therefore, the increase of adverse effects in pregnancy. Other studies have demonstrated the role of these cells through the inhibitory molecule leukocyte immunoglobulin-like receptor B4 (LILRB4), which

regulates the expression of Arg-1 and IL-10 through the SHP-2/STAT6 pathway, resulting in the dysfunction of dMDSCs, which could contribute to adverse outcomes during pregnancy due to *T. gondii* infection (<https://doi.org/10.1186/s13071-023-05856-4>). Recently, it was also described that IDO activation by mature cDC1s promoted a tolerogenic profile of inflammatory cDC2s through metabolic communication via their production of the tryptophan metabolite l-kynurenine. This fact is not mentioned in this document (doi: 10.1016/j.immuni.2022.05.013). This work is novel in that it analyses in greater detail the expression of IDO, which together with Arg1 and IL10 contributes to the immunosuppressive properties of dMSCs during pregnancy, and its influence on dNKs, which are one of the main populations at the fetal-maternal interface that determine the proper course of pregnancy. It seems that this will be an important piece of work for the scientific community and the development of new knowledge in this area, which is essential for understanding the major players during pregnancy. The results could also be used to understand the cell populations status at the fetus-maternal interface that are certainly involved with other pathogens that establish infection and the development of pathology associated with interference in the biological processes during pregnancy. However, the phenotypic characterization of dMDSCs is based exclusively on the expression of HLADR and CD33 and is therefore very preliminary. This analysis will have to be carried out more thoroughly in order to draw clearer conclusions about the role of dMDSCs in the experimental conditions tested (<https://doi.org/10.1002/cpz1.561>). This document lacks information about some items included in the reporting summary of nature portfolio, namely for statistical analysis, some of the conclusions and statements are not clear such as "line 27-28: ...but obviously decreased..."); the co-immunoprecipitation procedures are not clearly explained (line 306-307: what does WCL mean?); some phrases are not understood (Examples: line 66-67; line 83-84; line 93-94: "all immune cells interact with each other..."; repetition of information: line 96-100 repeating information from lines 89-91; is not referenced the clones used of monoclonal antibodies for the experimental procedures. Is not understood the parameters evaluated for the mice mental status; is referred about the expression of nuclear expression of p52

and RelB by western blot, but is not described the organelle isolation procedure in the document; is referred that recently a novel decidual (line 600) MDSCs referring to references from 2017; Overall, this document has some strong points, in that the study of the interconnection between dMDSCs and dNKs should play determinant role during *T. gondii* infection, and the adjacent inflammatory response at the fetal-maternal interface and how this response influences metabolism and the production of metabolites such as Kynurenine which is also important to pregnancy and influence the development of the fetus. At present, there is still controversy in the characterization of dMDSCs, so precautions should be taken in the conclusions taken from the data obtained. The phenotypical characterization used in this study include different profiles of dMDSCs that shared the same protein profile expression with other co-existing populations at the fetal-maternal interface such as dDCs.

Response: Thank you for your valuable suggestions. We have carefully modified the manuscript you mentioned.

1) In our study we used the enough numbers of human and mice to draw clearer conclusions about the role of dMDSCs in the experimental conditions tested *in vivo* or *in vitro* experiments.

2) We have revised “but obviously decreased” into “but protein level was significantly decreased” in line 27 in the manuscript.

3) We have added the detailed information about WCL information in line 299-300 in the materials and methods.

4) We have revised “extracellular” into “intracellular” in the manuscript line 66, revised “However, certain factors can stimulate SOCS3 to promote the production of SOCS3 and then the target proteins bound to SOCS3 were degraded by proteasomes” into “However, certain factors can stimulate the production of SOCS3 in cells and then promote the degradation of the SOCS3-bound target protein by proteasomes” in the manuscript line 82-84 and revised “All immune cells interact with each other” into “All decidual immune cells work together” in the manuscript line 93.

5) We have deleted repetitive information about “IDO is an intracellular heme-containing enzyme that can catalyze L-tryptophan into Kyn” in the manuscript line 98.

- 6) We have added the clones of monoclonal antibodies in the “Materials and methods”.
- 7) About mice mental status, we have removed this description from the manuscript.
- 8) We have added the protocol of nuclear protein extraction in lines 278-293 in the part of “Materials and methods”.
- 9) We have deleted the improper description “Recently, a novel decidual” in the manuscript line 627-628.
- 10) Thanks for your comments. A large number of MDSCs are found during normal pregnancy and MDSCs play critical roles for a balanced immunological tolerance at the maternal-fetal interface. Here, we aimed to analyze the immunosuppressive capacity of total MDSCs, focused on the changes of IDO in decidual MDSCs after *T. gondii* infection and the IDO downregulation on the function of dNK as well as the adverse pregnancy outcomes. Although researchers have attempted to identify specific marker, it is well appreciated that M-MDSCs cannot be distinguished from inflammatory monocytes on the basis of cell surface makers alone nowadays. In majority studies about MDSCs, Gr-1 and CD11b were used to define MDSCs in mouse and CD33⁺HLA-DR⁻ cells were defined as human dMDSCs. In addition, MDSCs were a heterogeneous population of immature myeloid cells derived from bone marrow, which included immature DCs, monocytes, macrophages, and granulocytes. Researchers paid more attention about their important immunosuppression function of MDSCs, which was not reflected by single population of cells. Based on the above reports, we gated the decidual MDSCs with CD33⁺HLA-DR⁻.

Reviewer 3

The present work considers the effect of *Toxoplasma gondii* on MDSCs cells and the collateral effect towards dNKs. It is an innovative topic, important for the understanding of dNK immunomodulation in the presence of infected MDSCs cells. I have some points of view that are not entirely clear to me.

Comment 1. In material and methods it mentions that they are marked “CD33-positive” to purify dMDSCs. Can you explain if this marked is sufficient to purify dMDSCs since this CD is expressed by myeloid stem cells, myeloblasts and monoblasts,

monocytes/macrophages, granulocyte precursors and mast cells, and classical neutrophils?

Response 1: Thanks for your comments. MDSC is a heterogeneous group of immature myeloid cells, which are the precursors of dendritic cells, macrophages and granulocytes (<https://doi.org/10.1016/j.immuni.2021.04.004>). Cell population of CD33⁺HLA-DR^{neg} and CD33⁺HLA-DR^{+/-} cells in human early pregnancy decidua have also been characterized to human decidual MDSCs (<https://doi.org/10.1111/aji.12492>). According to these references, we gated human decidual MDSCs as CD33⁺HLA-DR⁻. *In vitro* experiments, we used HLA-DR negative selection and CD33 positive selection for decidual MDSCs separation, but I did not write clearly in material and methods, thank you very much for your comments, we have revised that in line 217.

Comment 2. Likewise, you mention that dNKS are marked with “CD3-negative and a CD56-positive selection.” Can you explain if this mark is sufficient to purify dNK since they are markers that cross with other cells?

Response 2: It was known that the most human decidual NK cells in early pregnancy is labeled CD16⁻CD56⁺ (<https://doi.org/10.3389/fimmu.2017.00467>; <https://doi.org/10.1016/j.jri.2022.103475>; <https://doi.org/10.1016/j.pregthy.2013.04.062>). In most, only CD56-positive isolation to purify dNK cells in decidua tissue (<https://doi.org/10.1093/humrep/dez260>; <https://doi.org/10.1080/14767058.2020.1817369>). The CD56 antigen is expressed by most NK cells and a minor T cell subset (CD3⁺CD56⁺ NKT cells). Hence, to eliminate the interference of T cells, we first used CD3-negative isolation, and then sorted CD56-positive cells to get purified dNK cells. So CD3⁻CD56⁺ mark is sufficient to purify dNK cells.

Comment 3. Most of the graphs show very large deviations but and statistically significant. Furthermore, they do not mention the sample size used, both in mice and in human placentas. Therefore, I suggest checking if the statistical method used is the most appropriate for this work.

Response 3: Thanks for your comments, the data of human sample was varied greatly

due to different individuals, and the analysis method we adopted was the paired t-test, so some graphs showed large deviations, but indeed the statistical differences was significant between groups. And the number of human samples or mice were 6-8 individuals in each group. We have added the detailed information in all figure legends.

Comment 4. Results section 1st mentions “including extremely poor mental status” This statement is unclear and very subjective, and there is no context of how it was evaluated.

Response 4: Thanks for your suggestion, we have removed this statement from the manuscript.

Comment 5. Figure 4b presents different banding patterns with a stat3 inhibitor. My question is why it was not completely inhibited if it is an inhibitor of stat 3 phosphorylation?

Response 5: Stattic is a potent STAT3 inhibitor, which can inhibit STAT3 phosphorylation. Different concentrations have different effects on different cells. In our study STAT3 inhibitor was used to explore the effect of STAT3 in IDO expression of decidual MDSCs during *T. gondii* infection, so we select an appropriate concentration could inhibit STAT3 phosphorylation and avoid affecting the alive status. The preliminary experiments 10 μ M stattic treatment was used to detect the change of protein level. Meantime, STAT3 phosphorylation was not completely inhibited, but that was not affected our result.

Comment 6. Figure 6b They evaluate the effect of IDO -/- on kyn, it is observed that the uninfected kyn is lower in mouse placentas than in human placentas, my question is why this effect if in figure 2c, d; it is observed that there is more IDO in mouse placentas than in those isolated from humans.

Response 6: In the human HPLC experiment, the supernatant of collected human dMDSC (2×10^7 cells) was detected each group, while in the mouse experiment, the supernatant was detected from digested suspension of the placental tissue (100 μ g) in each group. It was represented the level of kyn in supernatant metabolized by IDO. In flow cytometry, it was represented the mean expression level of IDO in each cell.

Comment 7. In the Supplementary Fig. 1 section, you can see 2 populations, how did you consider that the two populations are dMSCs?

Response 7: MDSC is a heterogeneous group of immature myeloid cells, which are the precursors of dendritic cells, macrophages and granulocytes (<https://doi.org/10.1016/j.immuni.2021.04.004>). Human MDSCs are defined as CD33⁺HLA-DR^{-/low} CD11b⁺, and include two major subsets based on their phenotypic and morphological features: polymorphonuclear-like MDSCs (PMN-MDSCs) and mononuclear phagocyte-like MDSCs (M-MDSCs) (<https://doi.org/10.1016/j.jri.2021.103316>). While M-MDSCs express the myeloid marker CD33^{high}, PMN-MDSCs display CD33^{dim} staining (<https://doi.org/10.1038/ncomms12150>; <https://doi.org/10.1158/1078-0432.CCR-17-3726>; <https://doi.org/10.1126/scitranslmed.aae0482>). A study in decidua showed that dMDSC included low- and high- expressed CD33 positive subsets (<https://doi.org/10.3389/fimmu.2020.01345>). Our results were similar with this, which the low- and high- expressed CD33⁺ HLA-DR^{-/low} cells belong to tolerant decidual MDSCs.

Comment 8. Finally they demonstrated the increase to mRNA of IDO (IDO protein decrease) and SOCS3 in cells infected with *toxoplasma gondii*, but it is not discussed how the parasite could interact with the cell to obtain this.

Response 8: Thank you very much for your valuable comments. The detailed molecular mechanism of the interaction between parasite and dMDSC was still unknown. This might be a comprehensive and worthwhile topic which might be explored in further study. And we have added the information of discussion in lines 698-700.

Reviewers' comments:

Reviewer #1 (Remarks to the Author):

Thank you to authors for the many of the clarifications.

Only some minor critiques, some figures still do not have the relative values set at 1 which would still help in highlighting differences.

With some of the data, the authors are using unpaired t-test where ANOVA might be more appropriate where they have more than two comparisons.

Reviewer #2 (Remarks to the Author):

I have read the rebuttal letter and the revised version of the manuscripts, and it still lack a correct statistical analysis of the data (example Figure 1d: It is not possible to directly compare three groups using an unpaired t-test). Further and in my opinion flow cytometry data is not well analysed (is not done an analysis of single cells) and the phenotypic characterization of dMDSCs is based exclusively on the expression of HLADR and CD33 and is therefore very preliminary. in my opinion the article is not suitable for publication.

Reviewer 1

Thank you to authors for the many of the clarifications. Only some minor critiques, some figures still do not have the relative values set at 1 which would still help in highlighting differences. With some of the data, the authors are using unpaired t-test where ANOVA might be more appropriate where they have more than two comparisons.

Response 1: Thank you again for your valuable suggestions. We re-analyzed the figures (Figure 2b; 3c; 4b, c; 5b; 6c, e; 7a, b, e, f) to guarantee that they have the relative values set at 1 which help in highlighting differences. According to your suggestion, we also analyzed all data with ANOVA in figures where they have more than two comparisons (Figure 1d; 4a, b; 6a, b, d, e, g, h; 7b, c, d, e, f).

Reviewer 2

I have read the rebuttal letter and the revised version of the manuscripts, and it still lack a correct statistical analysis of the data (example Figure 1d: It is not possible to directly compare three groups using an unpaired t-test). Further and in my opinion flow cytometry data is not well analyzed (is not done an analysis of single cells) and the phenotypic characterization of dMDSCs is based exclusively on the expression of HLADR and CD33 and is therefore very preliminary. In my opinion the article is not suitable for publication.

Response 2:

(1) Thanks for your suggestion. At first, in figure 1d, the data analyzed by an unpaired t-test was to focus the comparisons between infected and uninfected groups, IDO^{-/-} infected and infected groups. According to your suggestions, ANOVA might be more appropriate where they have more than two comparisons. So, we re-analyzed all data in figures with ANOVA analysis and revised the statistical methods in all figure legends.

(2) Thanks for your reminding, flow cytometry data in our study have been re-analyzed by gating single cells in Figure 2c and the gating strategy of flow cytometry has been revised in Supplementary Figure 1. In fact, we compared the two before in human data and found that there was no significant difference in statistics after gating single cells. So flow cytometry data in our mice study was not analyzed by gating single cells. Thank

you for your valuable comments, we must be modify the gating strategy and analyzed all the flow cytometry data by gating single cells in the future. Although single cells were not ringed first in the flow gating strategy of mice, we were convinced of the final results were the same.

(3) Thanks for your constructive suggestion. As we known, MDSCs were still a not fully defined cell subset because it was a heterogeneous group of immature myeloid cells which are the precursors of various cells including dendritic cells, macrophages and granulocytes. So, the definitive phenotypic characterization of dMDSCs was still need to be further determined. Researchers paid more attention about their important immunosuppression function, which was not reflected by a pure population of cells.

Predominantly, the phenotype of human MDSC was reported to be CD33⁺ HLA-DR⁻/^{dim} cells (*Nat Commun*, 2016,7:12150; *Clin Cancer Res*, 2018,24(19):4834-4844; *J Reprod Immunol*, 2021,145:103316). In decidua, Bartmann et al. also characterized dMDSCs in human early pregnancy as new population with CD33⁺HLA-DR^{neg} and CD33⁺HLA-DR^{+/-} cells (*Am J Reprod Immunol*, 2016,75(5):539-556). In our present study, according to these references, CD33⁺HLA-DR⁻ cells were gated as human decidual MDSCs. Also, in our present study, CD33⁺HLA-DR⁻ were used as the phenotype of decidual MDSCs might be preliminary. In the further studies, we think your suggestion will be very helpful for using more definitive markers to recognize different subsets (eg. M-MDSCs and G-MDSCs) of decidual MDSCs. Thanks for your constructive suggestions.

Reviewers' comments:

Reviewer #1 (Remarks to the Author):

I am happy with the revised version of the paper

Reviewer #2 (Remarks to the Author):

The authors do not fully respond to the observations made in the previous review, namely the failure to re-analyse the flow cytometry data of mouse dMDSCs (in single cells) as they found differences in the re-analysis of human cells. Even though the document has been revised, there are some gaps in the wording of the text sent and the requests for changes to the text in revision 2 were not considered.

Reviewer 1

I am happy with the revised version of the paper

Response 1: Thank you again for your valuable recognition.

Reviewer 2

The authors do not fully respond to the observations made in the previous review, namely the failure to re-analyse the flow cytometry data of mouse dMDSCs (in single cells) as they found differences in the re-analysis of human cells. Even though the document has been revised, there are some gaps in the wording of the text sent and the requests for changes to the text in revision 2 were not considered.

Response 2: Thanks for your suggestion. In order to evaluate the degree of cytoadherence in mouse samples, we collected three samples (placentae and uteri) of normal pregnant mouse. The methods of single-cell suspension preparation and antibody labeling are exactly the same as those described in the manuscript. The single cells were gated by the strategy of flow cytometry as figures 1 and 2 below. The analysis results showed that the degree of cytoadherence was very low and the proportions of single cells were all > 95% (figures 1 and 2 below). At the same time, the results demonstrated that the gating single cells or not has little effect on the results of the proportion of mouse dNK cells and dMDSCs (figures 1 and 2 below). Also, after re-analyzing by gating single cells of human data in this study, we found that there was little statistical difference between the data coming from gating single cells and the data from non-gating single cells.

Sorry, we didn't set the parameter of FSC-H or FSC-W in flow cytometry and the single cells couldn't be gated in present data of mouse. Based on the above-mentioned analysis, we are sure that our current data is trustworthy even though single cells of mouse samples were not firstly ringed in the gating strategy of flow cytometry. Thank you for your valuable comments and kindly consideration, we must pay more attention to the gating strategy in the future.

Addition according to the suggestion of Reviewer 2, the manuscript has been revised again.

Fig. 1 Gating strategy of mouse dNK cells. **a**, **b**, and **c** are three independent samples. Single cells were gated (upper) or not gated (below) by FSC-A and FSC-H and dNK cells were then marked by CD3⁻CD122⁺CD49b⁻.

Fig. 2 Gating strategy of mouse dMDSCs. **a**, **b**, and **c** are three independent samples. Single cells were gated (upper) or not gated (below) by FSC-A and FSC-H and dMDSCs were then marked by CD11b⁺ Gr-1⁺.

REVIEWERS' COMMENTS:

Reviewer #2 (Remarks to the Author):

Dear authors,

I have considered and agreed to correct the new data and flow cytometry analysis.
The authors must make the following alterations:

line 43- " immunocompromised individuals like pregnancy women" This sentence is wrong and must be corrected.

Pregnant women are not considered immunocompromised in the classic sense, but their immune system works differently than those who are not pregnant.
Please read the paper doi:10.1111/j.1600-0897.2010.00836.x concerning this aspect.

Immunocompromised individuals correspond to: HIV+ or autoimmune and organ transplant patients.

Line 170: replace "mouse models" by "pregnant mice models"

Line 187: replace "Cell preparation of mice" by "sample cell preparation"

The text must be proofread by a native English speaker.

Reviewer 2

Dear authors,

I have considered and agreed to correct the new data and flow cytometry analysis.

The authors must make the following alterations:

line 43- “immunocompromised individuals like pregnancy women” This sentence is wrong and must be corrected.

Pregnant women are not considered immunocompromised in the classic sense, but their immune system works differently than those who are not pregnant.

Please read the paper doi:10.1111/j.1600-0897.2010.00836.x concerning this aspect.

Immunocompromised individuals correspond to: HIV+ or autoimmune and organ transplant patients.

Line 170: replace “mouse models” by “pregnant mice models”

Line 187: replace “Cell preparation of mice” by “sample cell preparation”

The text must be proofread by a native English speaker.

Response 2: Thank you for your valuable comments. We have carefully modified the manuscript you mentioned.

1) We have revised “immunocompromised individuals like pregnancy women” into “in special populations, such as pregnant women” in lines 42-43 in the manuscript.

2) We have revised “mouse models” into “Pregnant mice models” in line 500 in the manuscript.

3) We have revised “Cell preparation of mice” into “Sample cell preparation” in line 516 in the manuscript.

Thank you for your kindly consideration, the manuscript has been proofread by native English speakers.